# DeepDive: estimating global biodiversity patterns through time using deep learning

Rebecca B. Cooper [1,2] ✉, Joseph T. Flannery-Sutherland [3] & Daniele Silvestro [1,2,4] ✉

Understanding how biodiversity has changed through time is a central goal of evolutionary biology. However, estimates of past biodiversity are challenged by the inherent incompleteness of the fossil record, even when state-of-the-art statistical methods are applied to adjust estimates while correcting for sampling biases. Here we develop an approach based on stochastic simulations of biodiversity and a deep learning model to infer richness at global or regional scales through time while incorporating spatial, temporal and taxonomic sampling variation. Our method outperforms alternative approaches across simulated datasets, especially at large spatial scales, providing robust palaeodiversity estimates under a wide range of preservation scenarios. We apply our method on two empirical datasets of different taxonomic and temporal scope: the Permian-Triassic record of marine animals and the Cenozoic evolution of proboscideans. Our estimates provide a revised quantitative assessment of two mass extinctions in the marine record and reveal rapid diversification of proboscideans following their expansion out of Africa and a >70% diversity drop in the Pleistocene.

Changes in biodiversity through time reflect fundamental mechanisms of species diversification and extinction; estimating their dynamics is crucial to understanding the history of life. The fossil record provides empirical evidence on which to base estimates, offering insight to the processes of extinction, recovery, expansion and faunal and floral turnover, while setting the context in which drivers of biodiversity change are interpreted[1–5]. Fundamental questions in evolutionary biology such as whether or not there are global limits to biodiversity[6,7], or how biodiversity has evolved, shaped by environmental change, mass extinctions, and biotic interactions[8–12] rely on our ability to infer diversity patterns in deep time.

While the fossil record is the most direct evidence of past biodiversity dynamics, it is influenced by a plethora of preservation and sampling biases. These biases reflect variation in sampling and digitisation efforts, accessibility of fossil sites, intrinsic preservation potential of different organisms, habitats, and geographic regions and their geological history[13–16]. The result is that the fossil record is an incomplete sample of past biodiversity, plagued by temporal, spatial and taxonomic heterogeneities, which lead to an inferred mismatch between true and sampled diversity patterns[14,17,18].

Efforts to tackle this issue have resulted in a range of widely-used methods that estimate diversity trajectories through time while accounting for variation in sampling intensity. These include several rarefaction methods[19–22], maximum likelihood or Bayesian models based on Poisson sampling processes[23,24], and lower-bound richness extrapolators[20,25–27]. While these methods mostly focus on accounting for variation in preservation rates through time, they do not address variation in the geographic scope, temporal duration or environmental representation of sampling[28]. Thus, spatial and temporal heterogeneity of the fossil record still invariably hampers global biodiversity estimates even after sampling standardisation[29], and a recent analysis of the shallow marine fossil record found that spatial sampling heterogeneity accounts for 50-60% of changes in standardised richness estimates[14]. The increasing appreciation of the extent of these biases in

[1]Department of Biology, University of Fribourg, 1700 Fribourg, Switzerland. [2]Swiss Institute of Bioinformatics, 1700 Fribourg, Switzerland. [3]School of Geography, Earth and Environmental Science, University of Birmingham, Birmingham, UK. [4]Department of Biological and Environmental Sciences, Global Gothenburg Biodiversity Centre, University of Gothenburg, Gothenburg 413 19, Sweden. ✉e-mail: rebecca.cooper@unifr.ch; daniele.silvestro@unifr.ch

the fossil record calls for a shift in research efforts toward more spatially explicit studies of diversity through both space and time[17,30]. Recent efforts reflect this shift[24,26,31,32] and make progress towards reconciling an understanding of the impact of spatiotemporal biases with methods to combat these or with theoretical models to simulate plausible biodiversity patterns[33–35]. Yet, spatially explicit methods have mostly been used to estimate variation in regional diversity through time and among regions[14,24,29,36], without providing a direct solution to the problem of estimating change in global diversity. Taxonomic biases also remain unaccounted for in the current range of available methods despite widespread problems from variable preservation and sampling of taxa[18,37–39]. While some models can account for different preservation rates across lineages[24,40], they do not explicitly account for the effects these biases have on unobserved lineages.

Here we present a framework for estimating biodiversity through time from fossil data coupling a mechanistic simulation approach with inference based on deep learning. We assess the performance of our approach, named DeepDive (Deep learning Diversity Estimation) through extensive simulations, demonstrating its ability to accurately estimate diversity trajectories even in the presence of strong temporal, spatial and taxonomic sampling biases. We then use DeepDive trained models to estimate global biodiversity dynamics for two animal groups: marine animals from the Late Permian to Early Jurassic[24] and the mammalian clade Proboscidea[41].

## Results

### An approach to infer biodiversity through time
We developed a framework to estimate biodiversity trajectories consisting of two main modules: 1) a simulation module that generates synthetic biodiversity and fossil datasets and 2) a deep learning framework that uses fossil data to predict diversity through time (Fig. 1). The simulation module is designed to generate datasets that reflect our understanding of the processes of speciation, extinction, fossilisation and sampling. The simulator generates realistic diversity trajectories, encompassing a broad spectrum of regional heterogeneities (e.g. Supplementary Fig. 1). Simulated data also include fossil occurrences and their distribution across discrete geographic regions and through time, which are generated to reflect a wide range of spatial, temporal and taxonomic sampling biases (Supplementary Fig. 2). These data are used to train a deep learning model based on a recurrent neural network (RNN[42–46]) implemented in the second module. The RNN uses features extracted from the fossil record such as the

number of singletons or the number of localities per region through time to predict the global diversity trajectory. Biogeographic information detailed in the simulation module is reflected in these features. By training the model on many different datasets the parameters of the RNN learn the general properties of the fossil record and are optimised to predict biodiversity trajectories across different evolutionary scenarios and sampling biases.

The DeepDive approach also allows us to tailor the set of training simulations to specific empirical clades by adding temporal and biogeographic constraints as we demonstrate with two empirical studies. This means that we can inform our trained models based on a priori empirical knowledge that might not be directly observable in the fossil record of the clade of interest, such as changes in connectivity between landmasses or ocean basins known from geological records and models, or previously inferred mass extinction events. For example, custom training simulations for the Proboscidea-like model include a requirement to start with the origination of the clade in the time frame and 5 continental regions that are only allowed to be occupied by simulated species at the estimated times that the clade moved into those continents, and an end constraint of minimum three extant species. Changing biogeographies can be defined by altering simulation parameters e.g. by increasing dispersal connectivity when two regions merge or taxa migrate into a new region, or variable rates of dispersal and connectivity can be used. In custom training simulations for the marine data, simulations start and end with many taxa and have a probability of mass extinctions informed by the two major mass extinction events known in the time frame of the study. The distribution of parameters in the simulated datasets can be compared to those in the empirical occurrence data, to ensure the range of parameters that are expected based on the empirical data fall within the range the model has had the opportunity to learn from. In this way, it is possible to customise simulations, and thereby customise models trained on these data, to a wide variety of potential evolutionary scenarios while ensuring training sets are relatable to the empirical case. More details are provided in the Methods with parameters and notation summarised in Supplementary Table 1.

### Performance of DeepDive
We validated the performance of the DeepDive trained models through extensive simulations covering a wide variety of diversification scenarios, generating training sets to optimise the models and assessing their performance on independently generated validation

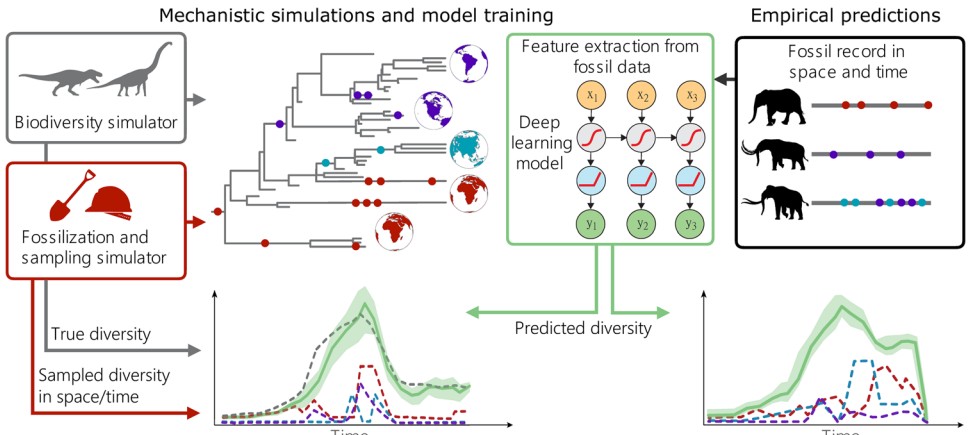

**Fig. 1 | Representation of the DeepDive workflow.** Biodiversity simulations generate global and regional diversity trajectories which are passed to a fossilisation and sampling simulator where these data are degraded to emulate the fossil record through time and across regions. The diversity trajectory and the simulated fossil record are then fed to a recurrent neural network (RNN). The RNN is trained to map the features of the fossil simulations back to the original simulated diversity curves from the biodiversity simulator in order to estimate biodiversity through time. The settings of the biodiversity and fossilisation simulators should be customised to reflect the spatial and temporal distributions observed in the empirical data (see e.g. histograms (Figs. 4–5)). Proboscidea silhouettes from phylopic.org.

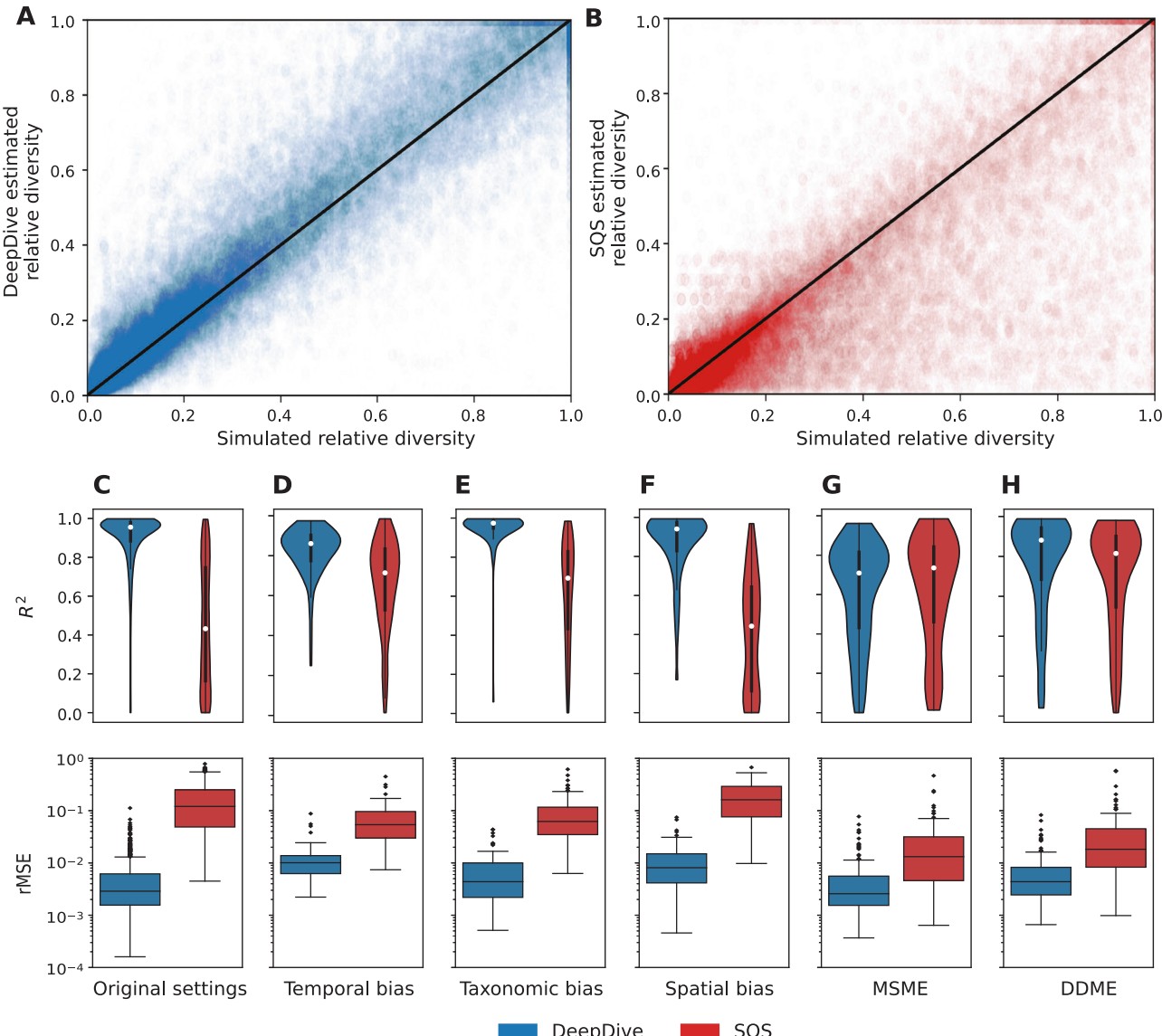

**Fig. 2 | The DeepDive approach (blue) can reduce errors and improve $R^2$ relative to SQS (red) under different bias conditions.** Accuracy of biodiversity estimations relative to simulated diversity for test datasets where (**A**) shows estimates made using DeepDive and (**B**) SQS at quorum level 0.6, in both cases the black line of slope 1 indicates the goal of these methods to make as close to a 1:1 estimate as possible. The variation in $R^2$ and relative MSE where (**C**) shows estimates on a test set generated under the same parameters as the training set, and for test sets generated to under different parameterisation to represent conditions of strong (**D**) temporal, (**E**) taxonomic and (**F**) spatial bias and for patterns that are rare in the training simulations (**G**) mass speciation and mass extinctions, (**H**) diversity dependence followed by mass extinction (see "Methods" for more details) for DeepDive and SQS. Data are presented as median values +/− the interquartile range, whiskers at 1.5 IQR. $n = 100,000$ (100 time bins × 1000 simulations). We note that the performance of DeepDive trained models substantially improved for settings (**G**) and (**H**) after re-training the model including sudden changes in speciation and extinction rates and diversity dependent processes in the simulations (Supplementary Fig. 7).

and test sets. We used a re-scaled mean squared error (rMSE) calculated after re-scaling both the simulated and the predicted diversity trajectories to a range between 0 and 1 and the coefficient of determination applied to the untransformed diversity ($R^2$) as performance metrics of the model. These allowed us to assess how accurately the predicted biodiversity matched the simulated trajectory, on a relative scale. Although we focus on relative diversity in our simulations, as this allows for fairer comparison with the subsampling approach of Shareholder Quorum Subsampling (SQS), the output of DeepDive quantifies absolute diversity through time. We therefore also report mean squared error (MSE) between the simulated and predicted diversity trajectories to quantify the accuracy of DeepDive.

We tested a range of model architectures to evaluate the optimal configuration (Supplementary Table 2) and found that most models perform similarly well within a range of validation MSE around 0.114−0.132 and test MSE between 0.197−0.229. This indicates that predictions are consistent across a range of parameterisations within our RNN framework. The predictions were accurate across a wide range of trajectory scenarios, in most cases closely matching the simulated ones (Supplementary Fig. 3 and Fig. 2A). To quantify the uncertainty associated with predictions, we included a Monte Carlo dropout layer[46] and made multiple predictions for each model. We combined the predictions from Monte Carlo dropout across different trained models to obtain 95% confidence intervals around diversity estimates. We found that while the predictions closely resemble the patterns of the simulated biodiversity values, simulated values are not in the 95% confidence intervals obtained through Monte Carlo dropout in a non-negligible fraction of time bins across the test set simulations.

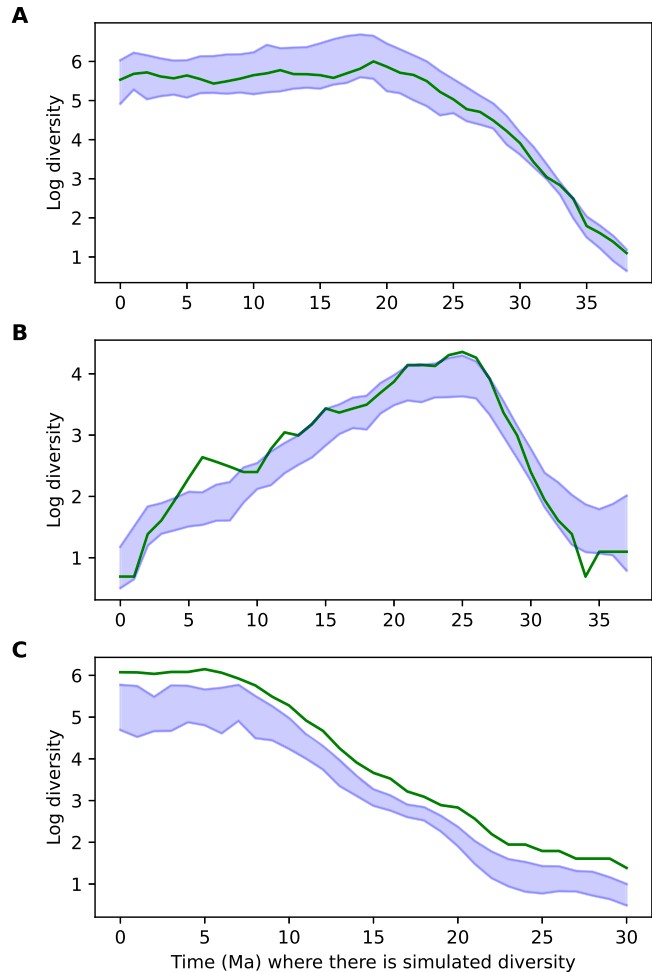

**Fig. 3 | Coverage of DeepDive predictions on simulated diversity curves.**
DeepDive estimates (blue) and simulated diversity (green) through time where
diversity exceeds zero demonstrating variable coverage where (**A**) shows the
maximum coverage 100%, (**B**) the median coverage at 66% and (**C**) the minimum
coverage of 0% of DeepDive estimates across 1000 test sets.

Specifically, the median coverage, i.e. the fraction of simulated values
included in the 95% interval, was 66% across test simulations (Fig. 3),
reflecting a tendency in Monte Carlo dropout to underestimate true
uncertainty intervals[47].

We evaluated how the performance of the DeepDive trained
models is affected by variation in the quality of the fossil record, which
we quantified as (1) completeness (fraction of species with at least one
fossil occurrence); (2) preservation rate (average number of fossil
records per lineage per time bin); (3) number of sampled species; (4)
the duration of taxa; (5) the duration of clades. DeepDive predictions
were most accurate, as expected, in more complete datasets, with high
preservation rates and more sampled species. Low error (rMSE < 0.01)
was found in most simulations with completeness exceeding 0.2 (i.e.,
where up to 80% of species were not sampled in the fossil record), with
increasing frequency of datasets with higher error at lower com-
pleteness levels (Supplementary Fig. 4A). We observed a similar trend
with preservation rates, with variation in rMSE being lowest for data-
sets with higher preservation (Supplementary Fig. 4B) with the mag-
nitude of error only increasing substantially above 0.01 in datasets
with fewer than ~200 sampled species (Supplementary Fig. 4C).
Whether a clade becomes extinct by the end of the time frame of the
analysis or remains extant has no substantial effect on the accuracy
(Supplementary Fig. 4D). Predictions tend to be more error prone in
datasets characterised by on average short-lived species, while we

found no clear relationship between clade duration and the MSE
scores (Supplementary Fig. 4E–F).

## Comparison to SQS

We compared DeepDive with SQS[19,27], one of the most widely-applied
methods for estimating diversity trajectories from fossil data[48–51]. Since
SQS estimates relative diversity, we used the rMSE and $R^2$ metrics to
compare their performances. On analysis, DeepDive trained models
outperformed SQS with lower relative median rMSE by more than one
order of magnitude and a higher median coefficient of determination
(0.958 vs 0.432) across test simulations (Fig. 2). Notably, DeepDive
estimation appears to be more robust to gaps in the data and to better
capture the overall biodiversity dynamics including both smooth and
sudden diversity changes (Supplementary Fig. 3). While both DeepDive
and SQS estimates were negatively affected by low levels of com-
pleteness and preservation rates (Supplementary Fig. 5), DeepDive
maintained substantially more accurate predictions across the entire
spectrum of sampling scenarios (Supplementary Fig. 4).

We tested five additional simulated datasets (100 simulations in
each case), which reflected strong conditions of temporal, taxo-
nomic, and spatial biases, and patterns of diversity change that were
not explicitly included in the initial training set: diversity dependence
followed by a mass extinction and simulations with multiple mass
speciation and mass extinction events, using both methods
(Fig. 2D–H). The coefficients of determination showed that temporal
variation of sampling rates can be effectively corrected for under
both methods, although DeepDive estimates were more accurate.
Taxonomic and, to a larger extent, spatial biases led instead to a
substantially higher discrepancy between the two methods. In the
case of strong spatial biases, the median $R^2$ exceeded 0.9 for
DeepDive, dropping to ca. 0.25 for SQS estimates. When patterns that
are rare in the training set are analysed the performance of the model
decreases to levels similar to SQS in terms of $R^2$, while still main-
taining substantially lower rMSE (Fig. 2G–H). However, the inclusion
of mass speciation and diversity dependence in the training set of a
re-trained model improves accuracy (Supplementary Figs. 6–7).
Inclusion of patterns such as diversity dependence followed by mass
extinction can also reduce errors around estimation of mass extinc-
tion events (Supplementary Fig. 8).

## Estimating diversity from empirical data

We apply the DeepDive approach to two empirical datasets of dif-
ferent taxonomic and geographic scope and use our predictions to
evaluate relative and absolute biodiversity changes through time.
The first is a genus-level dataset of marine animals spanning from the
Late Permian to the Early Jurassic comprising 71,386 occurrences
across 5312 genera of bryozoans, cnidarians, brachiopods, molluscs,
echinoderms, foraminiferans, arthropods, chordates, poriferans and
retarians (see "Methods"). The second is a species-level fossil dataset
of the order Proboscidea, which include modern elephants and their
extinct relatives, comprising 2104 occurrences across 180 species
(see "Methods").

We trained models based on simulated datasets, reflecting the
number and duration of empirical time bins in each dataset, and
including where possible diversity and spatial constraints (see "Meth-
ods" for more details). The features in the simulated datasets cover the
range of features observed in the empirical datasets (Figs. 4–5). After
evaluating the accuracy of the models (Supplementary Tables 3–4) we
estimated diversity trajectories for the two datasets.

Marine diversity declines between the Late Permian and the
earliest Triassic, with the loss of up to 58% of genera (Fig. 6) at the
time of the Permian-Triassic mass extinction (PTME), although lower
magnitudes are possible (mean 24% loss of genera). The number of
genera recovered in the Early Triassic and surpassed pre-PTME levels
in the Middle Triassic. Diversity declines more gradually through the

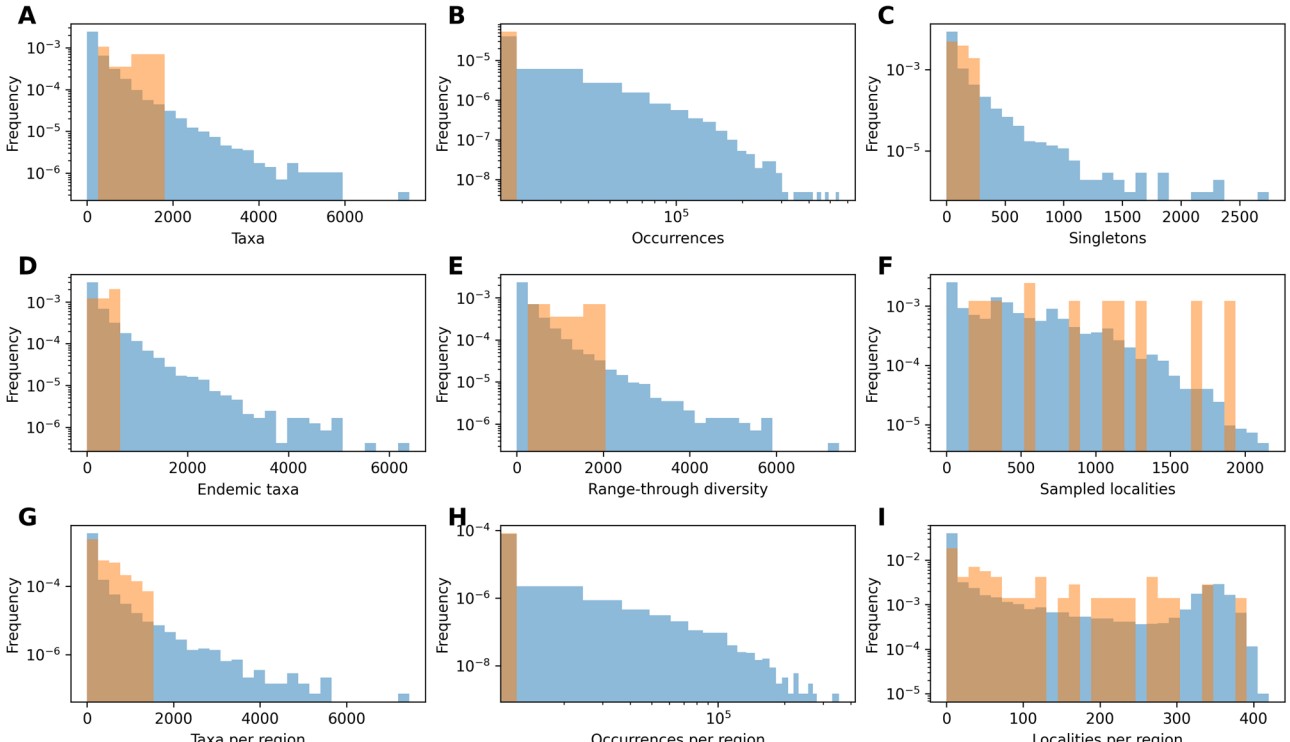

**Fig. 4 | Variation in DeepDive simulations relative to marine empirical data.** Histograms of the number of (**A**) taxa, (**B**) occurrences, (**C**) singletons, (**D**) endemic taxa, (**E**) the range-through diversity, (**F**) localities, (**G**) taxa per region, (**H**) occurrences per region, and (**I**) localities per region, per time bin where the range of values across 1000 Permo-Triassic marine-like DeepDive test simulations is shown in blue and empirical values from the uncorrected fossil dataset in orange. Features that describe the dataset are contained within the range of simulated values.

Late Triassic, followed by a sharper decline across the Triassic-Jurassic boundary with the loss of as many as 66% of genera (mean loss of 42% genera).

We estimated the diversification history of the proboscidean clade since its origin in the early Cenozoic (66–60 Ma) and inferred a gradual diversity increase through time leading to an estimated species richness of 10-20 species by the start of the Miocene (Fig. 7). In the Early Miocene (23–15 Ma) we detected a steep increase in diversity, leading to as many as 35–78 contemporary elephant species roaming Earth in the Middle to Late Miocene. Diversity remains relatively high and variable in the Pliocene and the start of the Pleistocene before crashing during the Pleistocene with 10–27 species at the end of the study interval (a loss of on average 65% species, maximally 87%, minimally 32%). In the light of this reconstruction, the modern diversity of three species results from over 70% species loss since the Pleistocene (2.58 Ma) (maximal loss 89%, mean 85%).

## Discussion

Recent research has shown that current methods to estimate biodiversity through time fail to account for biases in the data that are likely widespread across all fossil datasets, with severe impacts on the reliability of the predicted patterns[14,29]. Yet, the fossil record remains the most direct imprint left by the unfolding of life on Earth and computational methods can help us toward a more realistic interpretation of these data[52].

We present an approach for studying macroevolutionary changes in diversity from fossil data at large spatial scales, including globally, while accounting for temporal, taxonomic, and spatial sampling biases. Simulation-based model training provides us with the opportunity to capture complex biases in the fossil record that are difficult to implement in alternative available methods, which are primarily intended to correct for variation in sampling intensity through time.

Indeed, although methods like SQS are widely used for standardising diversity estimates, their intended purpose is to standardise samples to equal sampling intensity or completeness; they are not designed to control for variation in the scope of the accessible sampling universe, which is of central importance when estimating global diversity.

The deep learning models we use for inference allow connections to be made without attempting to explain how biases translate to a result, therefore making fewer prior assumptions and incorporating the potentially complex interactions between sampling efforts, true diversity, and their variation within and across datasets. Any prior assumptions made are parameterised explicitly in training simulations and must be justified, therefore the framework can potentially be used to assess impact of different sets of assumptions on biodiversity inference by generating models and test sets under different parameterisations. We found our model to be accurate across a wide range of diversity and sampling scenarios and to strongly outperform the most widely used current approach across simulations, indicating that this combination of mechanistic modelling and deep learning is worth the additional computational burden required compared with most of the existing alternative methods. DeepDive can produce accurate predictions with more depauperate fossil records, fewer sampled species, and at lower levels of completeness and preservation rates compared with alternative approaches (Supplementary Figs. 4–5). The accuracy of DeepDive estimates is highest when the true diversification scenario is present in the range of simulated training scenarios, while it decreases when diversification dynamics are absent or extremely rare in the training simulations. In both cases, relative MSE remains lower than with SQS estimates (Fig. 2G–H).

Unlike other unsupervised approaches[19,23,24], the method presented here uses mechanistic generative models within a supervised learning framework to estimate diversity through time. The approach capitalises on the fact that generating simulated data under complex

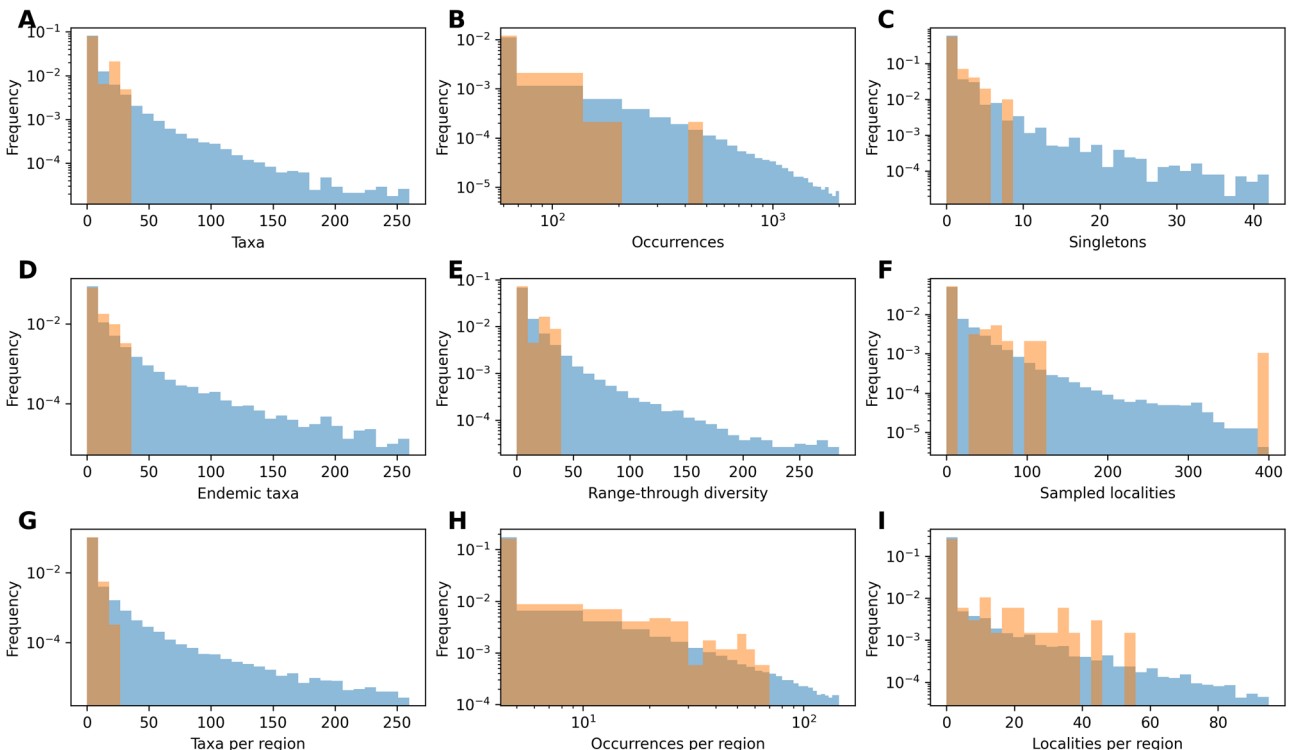

**Fig. 5 | Variation in DeepDive simulations relative to proboscidean empirical data.** Histograms of the number of (**A**) taxa, (**B**) occurrences, (**C**) singletons, (**D**) endemic taxa, (**E**) the range-through diversity, (**F**) localities, (**G**) taxa across per region, (**H**) occurrences per region, and (**I**) localities per region, per time bin where the range of values across 1000 proboscidean-like DeepDive test simulations is shown in blue and empirical values from the uncorrected fossil dataset in orange. Features that describe the dataset are contained within the range of simulated values.

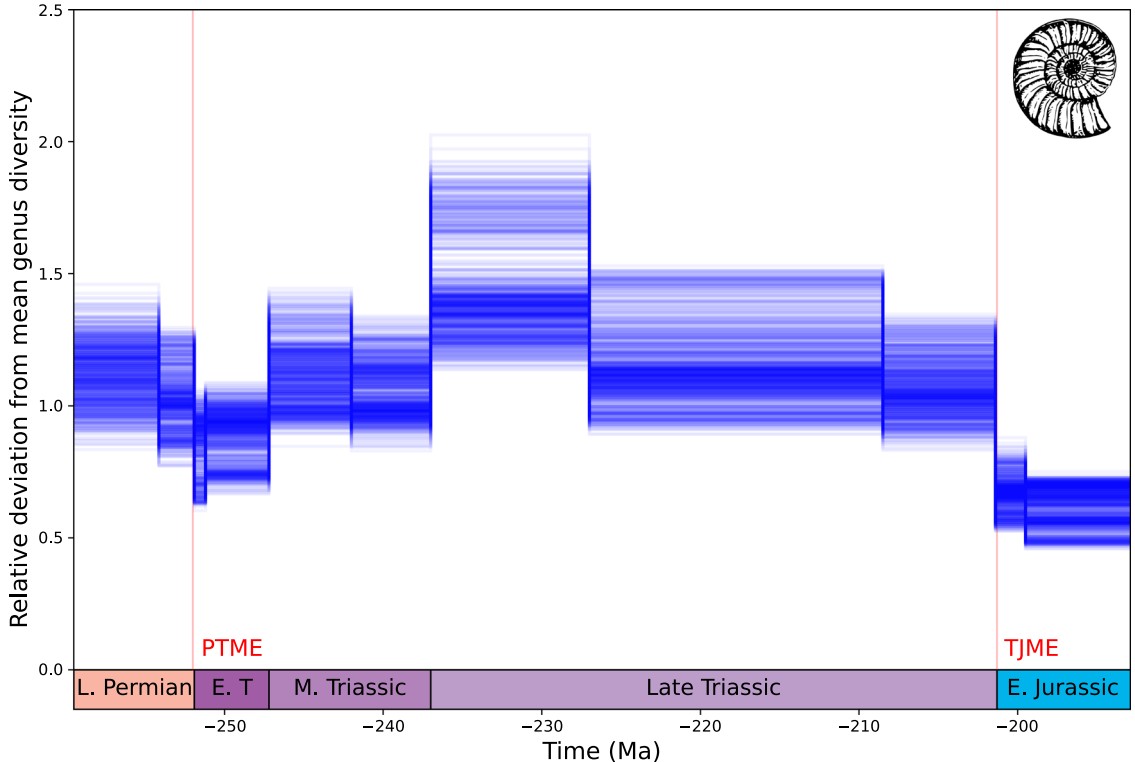

**Fig. 6 | DeepDive estimated diversity for Permo-Triassic marine genera.** Diversity declines at the Permo-Triassic mass extinction (PTME) but appears to surpass pre-PTME levels in the Middle-Late Triassic. Diversity declines in the Late Triassic before more substantial diversity losses around the time of the Triassic-Jurassic mass extinction (TJME). Previously inferred extinction events marked in red. Silhouette from https://freesvg.org.

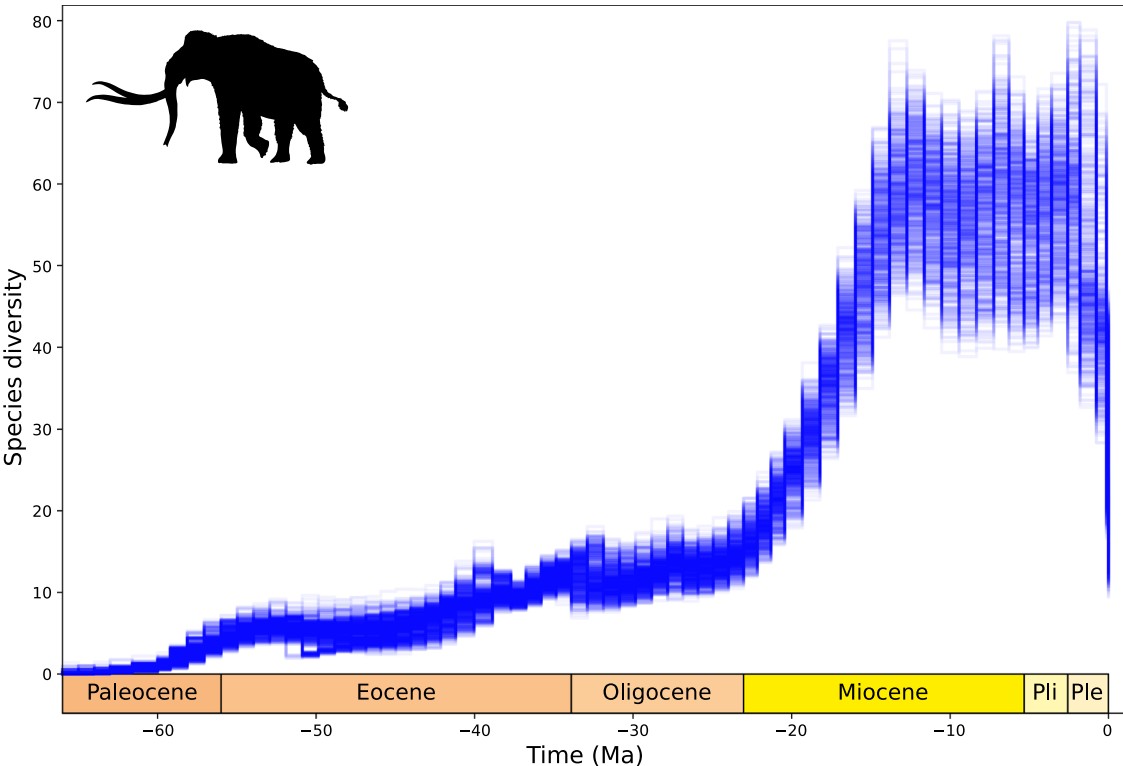

**Fig. 7 | DeepDive estimated diversity for Proboscidea.** There is an estimated gradual increase in the number of species during the Paleogene followed by more rapid accumulation in the Early Miocene, followed by an abrupt loss of biodiversity in the Pleistocene. Silhouette from phylopic.org.

scenarios of species diversification and fossil preservation in time and space is computationally less challenging than deriving and optimising the likelihood of the data under the same scenarios. A similar concept is also at the basis of other methods for inference such as approximate Bayesian computation, where simulations are used to bypass the calculation of the likelihood of the data given a model and its parameters[53–55]. The flexibility of the implemented generative models used here is demonstrated by the wide spectrum of biodiversity trajectories and preservation and sampling biases our model is able to reproduce (Supplementary Fig. 3). Because DeepDive uses a supervised learning framework, the accuracy of its predictions rely on the fact that the training set encompasses a sufficiently wide range of scenarios such that the empirical data we analyse fall within this range. Thus, there is a risk that a trained model generates erroneous predictions when faced with input data that differ substantially from the simulated training data[56]. This could occur if, for instance, a particular scenario like a mass extinction or a specific biogeographic sampling bias were present in the empirical dataset but not in the training set. This rationale was the basis for of our implementation of a simulation module that can capture a very wide range of scenarios, which indeed characterise the resulting datasets (Supplementary Fig. 3). We also showed that our framework lead to a substantial overlap between the features of the simulated datasets and those of empirical datasets (Figs. 4–5), thus suggesting that the training set has the breadth required to encompass the observed fossil patterns. The simulator needed to be highly computationally efficient, to allow for a reasonably fast generation of thousands of datasets. Thus, our simulations take a number of shortcuts by not modelling explicitly biotic interactions and simplifying biogeography to a predefined number of discrete geographic regions. Bioregions could be informed by using methods such as network community detection[57,58] or minimum spanning trees[24,29], to ensure they reflect the biogeographic structure of the data. While there cannot be certainty that all possible diversification and sampling scenarios are accounted for in any model or training set,

the generative models implemented in DeepDive can be extended to accommodate other patterns and biases in addition to those considered here. This could potentially include individual-based time and space specific simulations such as those presented in the gen3sis software[34], with the caveat that more complex simulations will increase the computational cost of the model.

Machine learning methods have recently gained increasing traction in macroevolutionary research. For instance, unsupervised machine learning has been used to estimate species co-occurrence probability and turnover through time[59]. Supervised models, such as gradient boosted trees, random forests and neural networks, have been used to predict the selectivity of mass extinctions based on traits and last appearance dates observed in the fossil record[60–64], and to predict the effects of extinctions on food webs based on modern species observations[65]. Machine learning approaches are also gaining traction in morphological studies[66] and in tasks such as determining taxonomy from images[67,68]. Supervised models have also been used to predict changes in paleo-vegetation based on expert-based interpretation of plant micro-fossils[69]. We chose to use recurrent neural networks in our implementation as they provide a direct way to learn from time-series, in this case fossil features through time, accounting for temporal dependencies in the data, and output another time series (diversity through time). We note, however, that our modular implementation would easily allow for the testing and comparison of different statistical or machine learning models. These could include predictive methods adapted for time series analysis, such as autoregressive models, generalised additive models, regression trees and random forests[70,71].

Training supervised learning models require knowledge of a ground truth, i.e. a labelled dataset on which the model can be trained on to make predictions[56]. However, in the case of palaeodiversity, and in much of macroevoutionary research, e.g. phylogenetic inference, the ground truth is unknown and cannot be experimentally verified. This means that both unsupervised and supervised models cannot be

independently validated beyond assessing their performance with simulations[24,72]. Unsupervised models rely on probabilistic models that describe a process mechanistically (e.g. speciation, extinction, and sampling in the diversity estimators implemented in TRiPS[23] and mcmcDivE[24]). Recent studies have shown that the same mechanistic processes can be used—in the absence of ground truth data—to generate datasets to train supervised deep learning models, achieving similar or better performance compared with probabilistic inference methods. These have been used to estimate age-dependent extinction[73], population genomics parameters[74,75] and birth-death rates in a phylogenetic context[76]. Our approach shows that coupling mechanistic generative simulations with supervised learning models can generate robust estimations of paleodiversity, strongly out-performing alternative methods across simulations especially in the presence of spatial biases.

Our empirical results suggest that marine diversity declined from the Late Permian into the Early Triassic across the Permo-Triassic mass extinction (PTME)[77] losing up to 58% of genera, comparable with some previous estimates[78] but with potentially lower magnitudes of diversity loss than previously hypothesised (mean loss 24%)[78–80]. Lower esti-mates may result in part from attempts to account for the regional variability of the mass extinction event, indeed heterogeneous responses have been documented across the globe in data analysed here[24]. From the PTME onward genus diversity accumulates to its highest point in the study interval at the start of the Late Triassic before entering long term decline. Diversity declines through the Middle and Late Triassic, prior to the Triassic-Jurassic mass extinction (TJME)[81] and our analyses indicate a sharp drop around that time. The global signal of extinction recovered is strong despite regionally het-erogeneous impacts of the event[24]. Patterns of genus-level diversity loss occurring in the lead up to and at the TJME are broadly consistent with previous hypotheses on global biodiversity changes at this time[81–83]. However, we note that comparing magnitudes of diversity loss estimated at the PTME and the TJME are at odds with what is expected from the literature. The exact magnitude of mass extinction events may be difficult to compare due to massive differences in sampling between the Late Triassic and Early Jurassic. Further, tests of the DeepDive trained models have shown that we expect slightly ele-vated errors after mass extinctions (Supplementary Fig. 8) and there-fore we must be careful when interpreting our estimations at the PTME and TJME.

General diversity trends found in our analysis of the proboscidean dataset are broadly consistent with previous hypotheses[41], as expected given the estimated elevated levels completeness of their fossil records[84]. Yet, our estimates reveal that many more species of the elephant family might have existed that have not been (yet) recovered in the fossil record. The low initial species-level diversity undergoes a gradual increase from the Paleocene to the end of the Oligocene, suggesting that diversity fluctuations observed in previous estimations might result from sampling biases rather than changes in diversifica-tion dynamics[41]. Species richness begins to accumulate at greater levels from the start of the Miocene, following the expansion of ele-phants out of Africa facilitated by the formation of the Gomphother-ium landbridge around 21–19 Ma[85]. This rapid diversification event is associated with increased ecomorphological exploration, niche parti-tioning and reduced competition among proboscideans during this time[41]. Our results potentially indicate a peak in genera level diversity shortly following the Mid-Miocene Climatic Optimum (17-15 Ma)[86,87]. Diversity plateaued throughout the latter part of the Miocene, Pliocene and beginning of the Pleistocene (2.58 Ma) with an estimated standing diversity of 37 to 75 elephant species during the Middle-Late Miocene, exceeding the maximum diversity recovered in other analyses and potentially representing a global carrying capacity for these mega-herbivores. Between 10 and 27 species are recorded at the end of the study interval after a Pleistocene diversity crash, broadly agreeing with

previous findings[41,88] and demonstrating the extraordinary magnitude of megafauna diversity loss in the recent past. This reduction in diversity eventually leading to three extant species has been attributed to global cooling with greater and more frequent fluctuations in temperature[89], reduced productivity and ecological disturbance[90], anthropogenic interactions[88,91,92] or possibly some combination of these changes[93].

The estimation of variation in global biodiversity through time is crucial to understanding the evolution of life on Earth and yet remains a highly challenging task given the temporally, spatially and tax-onomically biased nature of the fossil record[17], and because it is impossible to experimentally validate the accuracy of the estimations with real world data. The method proposed here develops an approach to the estimation of biodiversity through time by combining biological models of evolution with fossilisation and sampling processes, leading to a realistically incomplete and biased fossil record. These generative models are then integrated within a deep learning predictive model which produces accurate diversity estimates based on simulations and realistic predictions for empirical clades. We think this approach paves the way for a re-assessment of the major transitions and biodiversity dynamics across many lineages in the history of life.

## Methods
### A biodiversity simulation framework
The initial simulation has three main goals: to simulate diversity tra-jectories using a birth-death process, to simulate the biogeography of taxa (where they occur in a defined number of discrete geographic regions given a number of fossil localities), and to degrade the data to simulate the fossil record via the implementation of preservation and sampling biases. Our simulation framework is designed to allow for maximum flexibility in how the processes are modelled, such that the simulations capture the broadest spectrum of scenarios of varying diversification, extinction and sampling, while still allowing for highly efficient modelling to enable the generation of thousands of datasets.

Species origination and extinction times are generated using a stochastic birth-death process[94,95]. We use time-forward simulations in which the time of origin ($t_O$) is randomly drawn from a uniform dis-tribution, here set to $t_O \sim \mathcal{U}[30,100]$. Speciation and extinction rates are drawn independently from a uniform distribution $\mathcal{U}[0.05,0.5]$, i.e. reflecting a range of values commonly found in empirical fossil datasets[96,97] and are allowed to vary through time using a piece-wise constant model where the number of rate shifts are drawn from a Poisson distribution, Poi(4), and times of rate shifts drawn indepen-dently for speciation and extinction from a uniform distribution $\mathcal{U}[t_O,0]$, where 0 represents the present. In addition, there is possibility of a mass extinction occurring within a time bin with probability here set to $p_{ME} = 0.01$ per million of years. We constrained the incidence of a mass extinction to occur only if at least 10 or more lineages are present prior to that time bin, to prevent simulating clades from going extinct before accumulating a sufficient number of species for followup ana-lyses. A mass extinction is simulated by setting the extinction prob-ability for that time bin to $\mu_{ME} \sim \mathcal{U}[0.8,0.95]$, consistent with empirical estimations (e.g. 90% species loss for the Permian-Triassic extinction[77], 30–80% species loss for the Triassic-Jurassic extinction[78]), while events of smaller magnitude can still occur through the piece-wise rate var-iation. Depending on the rate dynamics and on the stochasticity of the simulation, the clade may or not have extant descendants. We con-strain the simulated datasets to a total diversity between 100 and 5000 species by discarding simulations not meeting or exceeding this target. These lineages are then counted across a set number of time bins of arbitrary size (here we use 100 time bins of 1 Myr) to generate the simulated global biodiversity trajectory. The birth-death settings used here ensured that a wide range of diversity trajectories were generated (Supplementary Fig. 3). Additionally, the settings can be customised to reflect empirical expectations related to specific

datasets, for instance conditioning on survival for clades with extant species or changing the expected frequency and magnitude of mass extinctions (see Empirical Analysis).

Simulated species are assigned to a predefined number of discrete biogeographic regions to simulate their biogeographic distribution. This allows us to apply spatially non-homogeneous variation in sampling, which has been shown to play an important role in diversity patterns in the fossil record[14,17,29,98]. Assignment of geographic ranges to each species depends upon three sets of parameters: the relative size of the region, the probability of origination at the region, and a distance matrix describing the connectivity among regions.

Relative region sizes modulate the relative diversity of each region as an approximation of a regional carrying capacity. The relative size of each region $d_a$ is drawn from a Dirichlet distribution

$$\{d_1 \ldots, d_A\} \sim \text{Dir}(\alpha \ldots, \alpha) \times A \quad (1)$$

where $A$ is the number of regions and the concentration parameter ($\alpha$) describes how similar the values drawn for size will be across regions (e.g. $\alpha >> 1$ leads to regions of similar size). The Dirichlet values are then multiplied by the number of regions ($A$) to re-scale the mean relative size across regions to 1. Alpha can be either initialised with a fixed value or drawn from a random distribution. The relative carrying capacity of a region $i$, which we indicate with $\kappa_i$, is further determined by parameter $k$, linking region sizes to their probability to host a species such that $k >> 1$ increases the difference between large and small regions while $k << 1$ reduces the link between region size and their probability to host species. The $\kappa$ values can be interpreted as proxies for relative carrying capacity of each region and sampled from a Dirichlet distribution

$$\int_a^b \{\kappa_1 \ldots, \kappa_A\} \sim \text{Dir}(d_1 \times k \ldots, d_A \times k) \quad (2)$$

Species geographic ranges are established by first drawing one initial region where a species is present and then by sampling the presence of the species in other regions based on a distance matrix. We indicate with $R_s$ the initial region of species $s$. The initial regions are generated as a function of relative carrying capacity such that species are more likely to be initialised within larger regions (depending on the parameter $k$), thus introducing a species-area relationship, based on a multinomial distribution

$$R_s \sim \mathcal{M}(\kappa_1 \ldots, \kappa_A) \quad (3)$$

Our simulation additionally allows for the carrying capacities to vary over time. In particular, given a vector of carrying capacities at time 0, $\kappa_1^{(0)}, \ldots \kappa_A^{(0)}$, the carrying capacities at time $t$ are defined as:

$$\kappa_1^{(t)}, \ldots \kappa_A^{(t)} = S\left(\log\left(\kappa_1^{(0)}\right) + c_1 \times t, \ldots, \log\left(\kappa_A^{(0)}\right) + c_A \times t\right) \quad (4)$$

where $c$ is a region-specific slope defining how the carrying capacity is varying over time, and $S(v_1, ..., v_A)$ is the SoftMax function:

$$v_i = \frac{\exp(v_i)}{\sum_j(\exp(v_j))} \quad (5)$$

Thus, in our simulations regions can have different relative carrying capacities and these carrying capacities can vary through time (Supplementary Fig. 9A). The flexibility of this parameterisation reflects the fact that a regions carrying capacity does not only depend on its size but on other factors too, such as productivity or climate, which can change through time[7,33].

Given the initial region of a species ($R_s = i$), we then sample whether the species also occurs in other regions as a function of a species-specific dispersal rate $w_s$ and a vector defining the relative distances ($\delta_{ij}$ for $j \neq i$) between $i$ and all other regions. After drawing the dispersal rate from a Weibull distribution $w_s \sim W(\phi, \psi)$ and the distance matrix from a uniform distribution $\delta_{ij} \sim \mathcal{U}(0, \delta_{\max})$, the probability for species with initial range $i$ to also occur in region $j$ is defined as:

$$P(j|i) = \exp\left(-\frac{\delta_{ij}}{w_s}\right) \quad (6)$$

Thus, the probability of occurring in other regions increases with higher species dispersal rate and with lower distance to the initial region.

## Fossilisation

After generating the complete evolutionary and biogeographic history of a simulated clade, with species origination and extinction and their spatial distribution, we degrade these data to simulate an incomplete and biased fossil record. We designed the simulation of fossil data aiming to capture the wide range of sources of bias and incompleteness expected in empirical fossil datasets. Specifically, we introduce heterogeneity in the fossil sampling rates across regions and taxa, and through time.

First, we draw for each region and time bin a number of fossiliferous localities within which fossil occurrences may be sampled. The expected number of localities in a region in a time bin is modelled as a function of a region-specific rate, the region size, a time-dependent rate and random effects. This approach provides a flexible framework to incorporate a wide range of plausible sampling biases. For each region $a \in \{1, ..., A\}$, we draw a region-specific rate from a gamma distribution with shape and rate parameters $\alpha_r, \beta_r \in \mathbb{R}^+$ multiplied by the relative size of the region $d_a$:

$$r_a \sim \Gamma(\alpha_r, \beta_r) \times d_a \quad (7)$$

Thus the expected number of fossiliferous localities in a region is a function of its size (larger regions will tend to harbour a larger number of localities), while the gamma distributed multiplier allows regions to differ in their intrinsic potential for fossilisation per unit of area. We then model sampling heterogeneity through time based on a pattern randomly selected from two implemented patterns: a consistent trend toward improved sampling through time with random variation, and a piece-wise constant model with random shifts. In the former pattern the mean sampling rate at time $t$ is defined as

$$E(\log(q_t)) = \log(q_0) + \zeta \times t \quad (8)$$

such that $q_0$ is the rate at the present ($t = 0$) and $\zeta \leq 0$ implying constant or decreasing rates moving back in the past. Log rates through time (Supplementary Fig. 9B) are then drawn from a normal distribution centred in $E(\log(q_t))$ and with standard deviation $\sigma \in \mathbb{R}^+$ modelling random variation around the general trend determined by the slope $\zeta$:

$$\log(q_t) \sim \mathcal{N}\left(E(\log(q_t)), \sigma\right) \quad (9)$$

The second pattern of heterogeneity through time is generated through a number of preservation rate shifts drawn from a Poisson distribution, in our simulations set to Poi(4). The times of rate shifts were drawn from a uniform distribution and the preservation rates between shifts were sampled from

$$\log(q_t) \sim \mathcal{N}(\log(q_0), \sigma) \quad (10)$$

In addition to modelling regional and time-specific rate variation, we include the possibility of complete gaps in the fossil record determined by an absence of fossil localities at given times and

regions. We model the occurrence of a gap for each time bin $t$ and region $a$ as a random draw from a Bernoulli distribution:

$$z_{at} \sim \text{Bernoulli}(1 - p_{\text{gap}}) \qquad (11)$$

where the parameter $p_{\text{gap}} \in [0, 1)$ defines the probability of a gap, such that increasing $p_{\text{gap}}$ results in a higher frequency of gaps in the fossil record. Finally, we include region and time-specific random effects with median $\eta \in \mathbb{R}^+$

$$\varepsilon_{at} \sim \eta \times e^m \qquad (12)$$

where the multiplier $m \sim \mathcal{U}(\log(1/b), \log(b))$ adds an amount of stochasticity around the median determined by a parameter $b > 1$.

We combine all elements defined above to obtain the expected number of localities $\lambda_{at}$ for each region $a$ and time bin $t$

$$\lambda_{at} = r_a \times q_t \times z_{at} \times \varepsilon_{at} \times \Delta_t \qquad (13)$$

where the rate is multiplied by the duration of the time bin ($\Delta_t$) to account for the fact that the number of sampling localities increases with the duration of the temporal range considered. The number of fossil localities $l_{at}$ in a region $a$ in a time bin $t$ is then sampled from a Poisson distribution:

$$l_{at} \sim \text{Poi}(\lambda_{at}) \qquad (14)$$

After obtaining the number of potential fossiliferous localities and accounting for spatial and temporal biases, we determine which species (if any) are preserved in each locality, based on their geographic range and on species-specific preservation rates. These are assumed to be gamma distributed, with $\psi_s \sim \Gamma(m, \theta)$, defining the expected number of fossils of a species $s$ per locality, conditional on the species being present in the region and at the time the locality records. Based on the rate $\psi_s$, the probability of species $s$ to leave at least one fossil occurrence in a locality $p_s$ (i.e. the probability to be found in the fossil record of a locality) is defined as:

$$p_s = 1 - \exp(-\psi_s) \qquad (15)$$

Thus, for a given region $a$ at time bin $t$ the fossil record is $f_{at}$ drawn from a binomial distribution:

$$f_{at} = \{\text{Bin}(l_{at}, p_s) \text{ for } s \in S_{at}\} \qquad (16)$$

where $l_{at}$ is the number of localities and $S_{at}$ is the set of species living in region $a$ at time $t$.

**Parameter estimation**
The simulated fossil record is summarised in a number of features that are then used as the input of a deep learning model to infer variation in global biodiversity (the total diversity across all regions studied) through time. Features are estimated to describe the spatio-temporal distribution of the fossil data and are defined as summary statistics that can be computed for both simulated and empirical datasets. Specifically, we implemented one-dimensional features that include for each time bin the number of sampled species, occurrences, sampled localities, singletons (species sampled only in one locality), endemic species (species sampled only in one region), time bin durations and range-through diversity through time. We also included two-dimensional features describing the number of sampled species, localities, and occurrences per region and time bin.

We use a Recurrent Neural Network (RNN) to make predictions of diversity time series in a supervised learning framework. The RNN uses the features extracted from the fossil record as input and maps them

to a prediction of number of species through time (output). Thus the input is a matrix of shape ($f$, $t$), for $f$ features and $t$ time bins, and the output is a vector of length $t$. The model contains bidirectional 'long short-term memory' (LSTM[43,44]) units that learn trends at different time scales, and additional fully-connected layers can be specified in the model. The output at any given point in time (a biodiversity estimate in a given time bin) is predicted based on a set of normalised features specific to that time point (e.g. number of fossils, number of sampled species) as well as predictions made for the previous and following time bins. In this way the RNN accounts for the expected auto-correlation of biodiversity time series. The RNN is also provided with the simulated log-transformed biodiversity trajectory (using a $\log(x + 1)$ transformation) obtained from the birth-death simulation before fossil bias implementation for training. We use the mean squared error (MSE) between simulated and predicted diversity as the RNN loss function minimised during training.

Different model architectures are tested with 1 or 3 LSTM units, with 32 and 128 nodes respectively, and `tanh` activation function and `sigmoid` recurrent activation function. The LSTM units were either directly connected to the output layer or to a fully connected network with 2 hidden layers (32 or 64 nodes, with `ReLU` activation function (Supplementary Table 2). We used a `SoftPlus` activation function[99] to obtain positive values of log-transformed diversity. We implemented and trained the RNN model using the ADAM optimiser[100]. The training was based on 100,000 simulations, 20% of which were used a validation set, and used a batch size of 100. The validation MSE was monitored across the training epochs to prevent the model from over-fitting through a `patience` parameter set to 50 epochs.

After selecting the best model as the one with the lowest validation MSE, we used it to predict diversity trajectories across a test set of 1,000 datasets. To estimate confidence intervals around diversity predictions we used Monte Carlo dropout wherein a fraction of connections in the last fully connected layer is randomly removed in each iteration[45,46]. We performed 100 predictions for each dataset and summarised them calculating the mean and 95% confidence interval for each time bin.

We additionally evaluated the performance of the trained model with coefficient of determination $R^2$ and the relative MSE (rMSE; mean squared error computed on simulated and predicted trajectories re-scaled between 0 and 1), that are scale invariant and capture how accurately the relative changes in biodiversity were predicted. Coverage, the fraction of simulated values included in the 95% confidence interval, was also estimated. The test set was also analysed using the SQS approach[19,20] at quorum 0.6 using the software DivDyn[101] in R version 4.1.2[102] and we computed $R^2$ and rMSE for comparison with DeepDive estimates.

To quantify the ability of DeepDive to correct for different types of biases, we additionally simulated three testsets (each including 100 datasets) reflecting strong and specific biases (Fig. 2). Namely, we simulated a case with strong temporal bias by implementing a 100-fold variation in average sampling occurring across four time frames, with a 10-fold rate decrease between 33 and 22 Ma and a 10-fold increase between 2 Ma and the present. We then simulated a strong taxon bias by setting the species-specific preservation potential based on a bimodal distribution, such that the average variation across species reached one order of magnitude. Finally, we implemented a strong spatial bias by setting one of the five simulated regions to have an average sampling rate 10 times higher than the average, and another region to a rate 10 times lower, thus introducing a 2 orders of magnitude spatial variation in sampling. We note that these datasets were generated under different parameter settings compared with the training set.

We additionally simulated datasets under birth-death models that were not explicitly included in the initial training data. First we generated a test set where mass speciation (with speciation rates sampled from $\mathcal{U}(1,5)$) and mass extinction (affecting 70–90% of the species)

events occurred with with probability of 0.05 for each time bin. Second, we simulated datasets with diversity dependent speciation[103] based a carrying capacity sampled from $\mathcal{U}(50,200)$ and two mass extinction events fixed at 66 and 16 Ma. After analysing these data with the trained models, we re-simulated a training set that included these two birth-death settings and trained a new models. This allowed us to evaluate the improvement in the DeepDive predictions after the inclusion of additional diversification scenarios. Simulations with fixed times of mass extinction also allowed us to quantify the rMSE of the predictions as a function of time, to evaluate to what extent the predictions error varies at time of sudden diversity change.

### Empirical analyses with customised simulations

DeepDive was used to estimate biodiversity trajectories from two recent case studies in the literature: Late Permian to Early Jurassic marine fossils primarily sourced from the Palaeobiology Database (paleobiodb.org/)[104] and other datasets (see[105] for full reference list) as compiled by Flannery-Sutherland et al.[24], and Cenozoic proboscideans originally sourced from the New and Old Worlds database of fossil mammals (nowdatabase.org/) and the Palaeobiology Database and published literature as compiled by Cantalapiedra et al.[41]. Since the features used as input in the DeepDive trained models include counts of sampling localities, we used locality names when available in the raw datasets and otherwise defined localities as geographic points with a unique set of co-ordinates from which fossils of the same age range have been sampled. 100 replicates of age assignment are made for occurrences in each dataset by drawing from a random uniform distribution between age range of the locality which they belong to. We trained new models for each empirical dataset, incorporating in our simulations prior knowledge about the analysed biological systems. Specifically we tested models with 1–4 LSTM layers, and the addition or absence of a fully connected layer. After finding similar fit across models (Supplementary Tables 3–4) we decided to generate and aggregate predictions from the empirical datasets based on all models.

For the genus-level marine occurrences, the spatially standardised sampling regions from[24] were used to define five geographic regions in DeepDive: the South Panthalassic, North Panthalassic, Boreal, Tangaroan, West Circumtethys and East Circumtethys. 11 geological stages from the beginning of the Wuchiapingian (259.51 Ma) to the end of the Sinemurian (192.9 Ma) are set as the time bins. Custom training simulations that reflect the general properties of the Permo-Triassic marine fossil record were generated by providing the simulator with some a priori information. Probability of a mass extinction is set to 0.029 per my, equivalent to the expected two events that occurred in the 70 myr time span of the dataset, i.e. the Permo-Triassic[77] and Triassic-Jurassic[81] mass extinctions. The number of starting species at the beginning of the simulation is set between 100 and 1000, with no fewer than 100 species extant by the end of each simulation. This ensures that the simulations begin and end with many species still in existence, such that none of the simulated diversity curves reflect the origination or extinction of all taxa studied as is appropriate when studying truncated datasets. 150,000 training and 1000 test sets were generated.

For the species-level proboscidean dataset, geographic regions include Africa, Europe, Asia, North America and South America. 69 time bins averaging 1 ma in length are set spanning from the start of the Danian (66 Ma) to the end of the Holocene (0 Ma). All simulations in this case start from 1 species as the clade originates within the study interval, with minimum extant species constrained between 3 and 30 such that simulated curves do not reach extinction of all taxa or end below the number of living elephant species. The following geographical constraints were applied to the simulations: that elephants are allowed to occupy Africa for the full time study interval[41], that they arrive in Europe and Asia between 33.9 and 27 Ma[41], in North America between 20 and 16 Ma[41] and in South America between 5.3 and 0.8 Ma[106]. 150,000 training and 1000 test simulations were generated.

### Reporting summary

Further information on research design is available in the Nature Portfolio Reporting Summary linked to this article.

## Data availability

Data, simulations and trained models used in this study are available on a permanent repository at https://zenodo.org/records/10979237 (https://doi.org/10.5281/zenodo.10979237).

## Code availability

DeepDive is implemented in Python 3.10, relying on the Tensorflow v.2.8 library[107] for the deep learning optimisation and on the numpy v.1.22 (numpy.org) and scipy v.1.8 (scipy.org) libraries for numerical simulations. The code and scripts are available at https://github.com/DeepDive-project and at zenodo.org at https://zenodo.org/records/10979237 (https://doi.org/10.5281/zenodo.10979237).

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

## Acknowledgements

We thank Tiago B. Quental, Sara Varela, Amanda Gardiner, Catalina Pimiento, Fernando Blanco and Roger B.J. Benson for their feedback and comments on the DeepDive project and manuscript. RBC and DS received funding from the Swiss National Science Foundation (PCEFP3_187012). DS also received funding from the Swedish Research Council (VR: 2019-04739). JTFS received funding from the NERC GW4+ DTP (studentship S100065-138/123).

## Author contributions

R.B.C. and D.S. developed code and wrote the manuscript with input from JTFS. All authors approved the submitted version.

## Competing interests

The authors declare no competing interests.
