## [Peer Review File · Nature Communications]

Estimating global biodiversity patterns through time using deep learningReviewers' Comments:

Reviewer #1:

Remarks to the Author:

Cooper and collaborators present a new method to reconstruct paleodiversity curves based on occurrence information. Their approach uses a combination of simulations and trained neural networks to estimated absolute trajectories of taxonomic diversity over time. This is a welcomed innovation, because, for many decades now, paleodiversity estimates have been limited either to the use of residuals from correlations between sampling proxies and diversity 1,2, or, more commonly, different flavors of subsampling methods 3,4. Noteworthy, the proposed method incorporates potential geographic biases in sampling rates, a critical limitation of standar subsampling procedures5. Since the basis for diversity estimates comes from informed ad hoc simulations, the new method can accommodate virtually any macroevolutionary scenarios. Such simulated scenarios can incorporate shifting regional carrying capacities (e.g. climate-dependent species limits), shifting biogeographic connections, and changing sampling potential across time, geographic region and lineages. Together, these features should solve many of the crucial limitations of diversity estimates from the fossil record. I have some comments that should make the manuscript clearer for the broad readership of Nature Communications.

Since most of the readers would not dive through the methods section, I believe that the authors should bring a simplified version of some parts of the methods into the main text. In particular, some aspects related to the tailoring of the simulated trajectories to show the flexibility of the method. For example, how one could limit biogeographic connections to train the model for the proboscideans dataset. Also, the readers should get a better idea of the overall workflow, in particular how simulations and the analyses of the empirical data connect to each other. In the methods the authors explain what parameters are taken for the real data into the simulations. Then how the simulations are done and how the trained NN is run on the real data. Some brief version of this in the main text should help. Also, the pipeline in Figure 1 is not totally clarifying. It doesn't show how the real data is also used to inform simulations.

Regarding the input data, it is detailed that it takes the shape of a matrix, with several features computed for each temporal bin. From what I understand, no biogeographic information is used in this input, right? This is, biogeographic information from the real datasets is only used in the simulations, right? This should better clarified in the main text and methods.

These are just minor comments oriented to make the method more understandable for a broad audience. I believe that DeepDive will be a broadly used approach in future paleobiology and evolutionary biology research. A more precise assessment of deep-time biodiversity trends is key for a correct understanding of past and future responses of the biosphere to environmental disruptions.

Juan L. Cantalapiedra

Minor comments

Black line missing in Fig 6 and 7.

Line 150 - missing Figure number.

Line 275 – the oldest Asian proboscidean is dated at around 25 Ma, which pushes that age back a few million years.

Line 279 – maybe also mention the absolute age, for readers not familiar with geological periods.

Line 487 - indicate where this is shown in the results figures.

Table S4 caption – “Elephant-like” is a vague term. Elephantids are just one branch of the order Proboscidea. I would just say “Proboscidea”.

References

1. Raup, D. M. Species diversity in the Phanerozoic: an interpretation. *Paleobiology* 2, 289–297 (1976).

2. Lloyd, G. T. A refined modelling approach to assess the influence of sampling on palaeobiodiversity curves: new support for declining Cretaceous dinosaur richness. *8*, 123–126 (2012).
3. Raup, D. M. Taxonomic diversity estimation using rarefaction. *Paleobiology* **1**, 333–342 (1975).
4. Alroy, J. The Shifting Balance of Diversity Among Major Marine Animal Groups. *329*, 1191–1194 (2010).
5. Close, R. A., Benson, R. B. J., Saupe, E. E., Clapham, M. E. & Butler, R. J. The spatial structure of Phanerozoic marine animal diversity. *Sci New York N Y* **368**, 420–424 (2020).

Reviewer #2:

Remarks to the Author:

This paper represents an excellent contribution to the methods available for studying biodiversity in the fossil record. The DeepDive method is an important advance for our ability to circumvent biases that affect the scope of the sampling universe captured in the fossil record (geographic space, taphonomic biases, etc).

With existing methods, estimating global diversity would have been largely intractable, due to spatial bias in the fossil record. With DeepDive, however, we may now actually be able to estimate this parameter with some confidence, provided that the simulations used in the training step encompass the range of plausible scenarios that generated the empirical data.

I think that the paper should be published in Nature Nature Communications, but there are some relatively minor changes that need to be made, and I have some queries and comments on specific parts of the text. I have also annotated the PDF of the MS with suggested edits and comments.

General questions:

- Can the authors comment on whether the model can do a good job of inferring biodiversity trajectories outside of those encountered in the training data? How do we know when the breadth of the simulated training data isn't adequate? Will widespread application of this method result in disagreements between camps of researchers who differ in the way they think biases should be simulated in the training data, yielding different resulting curves? How could this scenario be resolved?
- Can the authors comment on how the model could be used over time intervals when palaeogeographies change? Can changing region definitions (including appearances and disappearances of regions) be accommodated by the DeepDive model?

Remarks and queries about particular sections of the text:

- 61–64: Are the inferred biodiversity trajectories biased or influenced by the distribution of trajectories recorded in the training data? I.e., will the model tend to infer the correct biodiversity trajectory if that kind of pattern is present but very rare in the training data? For example, if the vast majority of the simulated biodiversity histories are not diversity dependent, but a tiny fraction of them are diversity dependent, would the model do a good job of inferring diversity dependence if it was truly present in the empirical data?
- 76–77: Do independently-generated validation and test datasets have the same generating model parameters? Would the performance of the model be worse if test datasets were generated under different model parameters? Or is this not much of an issue for assessing the model's performance, more generally?
- 91–92: How strongly does the model want to infer temporal autocorrelation in diversity trajectories? I.e., does it generally tend to smooth out short-term variation, as in the smooth trajectory of S3A (I realise that the true trajectory was also smooth)? Or can it also correctly infer repeated and sudden short-term changes in diversity? I'm just wondering if the smoothness of the trajectories in the training dataset would result in inferred trajectories for empirical datasets being quite smooth by default, even if the true trajectory wasn't. On a related note, is the size of the confidence interval envelope for the inferred trajectory larger across all intervening bins when the empirical record has

large short-term fluctuations in sampling completeness, like in the examples of Fig. S3?

- Line 122 and elsewhere: Regarding the comparisons between DeepDive and SQS — I think that to be fair to SQS, and especially to prevent detractors from using your analysis as a reason not to use SQS in any context, it would be preferable to add some qualifications to these conclusions (as I mentioned in earlier comments on the PDF). SQS is a tool for standardising samples to equal sample completeness, which means that it's mainly geared towards correcting for uneven sampling intensity. It's not able to correct for variation in the size of the underlying sampling universe (spatial, taphonomic, taxonomic, etc.) — and neither can any method solely aimed at correcting for sampling intensity. DeepDive can obviously do both, but SQS is still an objective and reliable way to standardise diversity samples, given that certain assumptions are met. E.g., it would be good for standardising alpha diversity within localities across samples, or within spatial regions if these regions are fairly comprehensively sampled. If sampling is random and unbiased, and the size of the sampling universe is controlled for, then SQS does a good job of telling you how many species are found, on average, within a random sample of $q\%$ of the individuals in a population (e.g., 60% of the individuals at a quorum of 0.6, or 90% at a quorum of 0.9). I think that's biologically meaningful and informative.
- 247: Are there limits to how severe spatial biases can be before DeepDive cannot reliably estimate global diversity, and would we be able to recognise this situation? E.g., how would DeepDive cope with sampling regimes where entire continental regions are absent from the global fossil record for protracted intervals of geological time? If you look at how regions are sampled through the Phanerozoic in tetrapods (Close et al. 2020 Proc B) you'll see that there is a very heavy bias towards Europe and North America, with many other regions entirely absent from the record for long intervals. It might be useful to comment on these limits of DeepDive in the discussion.
- 311: Would you need to include (e.g.) diversity-dependent and early-burst models in the simulations in order to infer those patterns in empirical data? Even a wide range of diversity trajectories were generated in your simulations, are constant models with rate-shifts sufficient for all likely scenarios?

Reviewer #3:

Remarks to the Author:

Review of Cooper et al.

Summary

This study uses synthetic data for training a deep learning model aiming to infer the geological history of the global scale species richness. Specifically, the authors generate abstracted fossils records consisting of species occurrences distributed over arbitrary time intervals and predefined geographic units. These abstracted records are generated using a birth-death process and later filtered using both sampling and preservation filters to resemble the physical fossil record. The abstracted fossil records are summarized using different features, which are used to train a bidirectional long short-term memory neural network to predict species richness over time. Authors claim the proposed approach outperforms alternative methods under a wide range of preservation scenarios. Overall, this work provide a great alternative to overcome data shortages for machine learning model training in paleobiology, specifically for estimating biosphere-scale species richness, but with potential to test a number of hypothesis in metazoan macroevolution. I have no major issues with the data, methods, and conclusions, and I recommend publication with some minor revisions. Below I provide some suggestions that may help to improve the manuscript.

COMMENTS

On the simulated fossil records

Comment: I very much like the idea of using simulations to generate fossil records that capture biogeography structure. However, I would like to know why the authors assigned taxa to predefined bioregions to capture "biogeography of taxa". This approach may not reflect the biogeographic struture underlying the data.

Perhaps biogeographic structure can be extracted from the data without using a predefined number of discrete biogeographic regions. Recent studies show that unsupervised community detection on network representations of the same empirical data captures biogeography. So, I am wondering if it would be possible to use network community structure to capture biogeographic signal in both physical (=empirical data) and abstracted records (=simulated data). For instance, to compute the effective number of modules/communities at a given time.

After all, using a predefined number of discrete biogeographic regions that remains unchanged over time does not really capture temporal changes in spatial structure.

On the empirical studies

Comment: When estimating diversity from empirical data, the authors presented a case study that seems to include benthic and planktonic marine clades all together. Perhaps it is not a surprise that the simulated trajectory do not capture CPE. To test for CPE, it would be better to restrict the analysis to the clades that has been observed to be affected by such an event, otherwise this result remains somewhat speculative.

TJME vs PTME

Comment: The authors indicate that "patterns of genus-level diversity loss occurring in the lead up to and at the TJME are broadly consistent with previous hypotheses on global biodiversity changes at this time". However, the pattern (in Figure 6) also seems to indicate that TJME is stronger than PTME. Is that also consistent with previous hypotheses? The study provides a global measure of how good are the simulated trajectories. I am wondering if would be possible to assess that at the stage and epoch level.

Does DeepDive "outperform the most widely used current approaches"?

Comment: The study provides a global metric to evaluate the performance of the trained model. I am wondering if it would be possible to assess the performance at the stage and epoch levels that underlies the empirical data. We could learn a lot from that. For instance, what happens across known major geological events? Does the performance of the trained model drop? Does it recover gradually after? I believe that could be more interesting than the trajectory itself as it may capture major events over time and their relative impact on the species richness.

I hope you found these ideas useful,

- Alexis Rojas

Review 1

Cooper and collaborators present a new method to reconstruct paleodiversity curves based on occurrence information. Their approach uses a combination of simulations and trained neural networks to estimate absolute trajectories of taxonomic diversity over time. This is a welcomed innovation, because, for many decades now, paleodiversity estimates have been limited either to the use of residuals from correlations between sampling proxies and diversity 1,2, or, more commonly, different flavors of subsampling methods 3,4. Noteworthy, the proposed method incorporates potential geographic biases in sampling rates, a critical limitation of standard subsampling procedures⁵. Since the basis for diversity estimates comes from informed ad hoc simulations, the new method can accommodate virtually any macroevolutionary scenarios. Such simulated scenarios can incorporate shifting regional carrying capacities (e.g. climate-dependent species limits), shifting biogeographic connections, and changing sampling potential across time, geographic region and lineages. Together, these features should solve many of the crucial limitations of diversity estimates from the fossil record. I have some comments that should make the manuscript clearer for the broad readership of Nature Communications.

Since most of the readers would not dive through the methods section, I believe that the authors should bring a simplified version of some parts of the methods into the main text. In particular, some aspects related to the tailoring of the simulated trajectories to show the flexibility of the method. For example, how one could limit biogeographic connections to train the model for the proboscideans dataset. Also, the readers should get a better idea of the overall workflow, in particular how simulations and the analyses of the empirical data connect to each other. In the methods the authors explain what parameters are taken for the real data into the simulations. Then how the simulations are done and how the trained NN is run on the real data. Some brief version of this in the main text should help. Also, the pipeline in Figure 1 is not totally clarifying. It doesn't show how the real data is also used to inform simulations.

We thank the Reviewer for highlighting this and agree that this should be made clearer. Thus we have now relocated a summary of the relevant information regarding custom simulations for the empirical studies to the main text (lines 74-88). Additionally we have redrawn the workflow in Fig. 1 to include a representation of biogeography, and clarify that spatial data is included as part of the input to our DeepDive model and that the simulations can be customised to encompass the range of the features of the empirical data and e.g. known biogeographic or preservation patterns, in the figure caption.

Regarding the input data, it is detailed that it takes the shape of a matrix, with several features computed for each temporal bin. From what I understand, no biogeographic information is used in this input, right? This is, biogeographic information from the real datasets is only used in the simulations, right? This should be clarified in the main text and methods.

Biogeographic information is in fact used as part of the input data. The model uses both the spatial and the temporal distribution of fossil occurrences to make its predictions.

The computed features include time series per region e.g. the number of occurrences or the number of singletons for each region (among others). In this way biogeographic information is passed from simulations during both the model training phase of the workflow and the prediction phase, when these features are extracted from the distribution of occurrences in the empirical data to estimate biodiversity through time. We have now stated this important aspect of the analytical workflow in lines 62-64.

These are just minor comments oriented to make the method more understandable for a broad audience. I believe that DeepDive will be a broadly used approach in future paleobiology and evolutionary biology research. A more precise assessment of deep-time biodiversity trends is key for a correct understanding of past and future responses of the biosphere to environmental disruptions.

Juan L. Cantalapiedra

Many thanks for the insightful comments on our work, we addressed all points below in revising our manuscript.

Minor comments

Black line missing in Fig 6 and 7. **Caption updated**

Line 150 - missing Figure number. **Missing figure number added**

Line 275 – the oldest Asian proboscidean is dated at around 25 Ma, which pushes that age back a few million years. **This is compatible with the decision to simulate movement into Europe and Asia between 33.9-27 Ma, we rephrase to “following” the formation of the Gomphotherium land bridge to avoid any confusion (line 321).**

Line 279 – maybe also mention the absolute age, for readers not familiar with geological periods. **Ages added for the Mid-Miocene climate optimum and start of the Pleistocene.**

Line 487 - indicate where this is shown in the results figures. **Added figure references.**

Table S4 caption – “Elephant-like” is a vague term. Elephantids are just one branch of the order Proboscidea. I would just say “Proboscidea”. **Wording changed.**

References

1. Raup, D. M. Species diversity in the Phanerozoic: an interpretation. *Paleobiology* 2, 289–297 (1976).
2. Lloyd, G. T. A refined modelling approach to assess the influence of sampling on palaeobiodiversity curves: new support for declining Cretaceous dinosaur richness. *8*, 123–126 (2012).
3. Raup, D. M. Taxonomic diversity estimation using rarefaction. *Paleobiology* 1, 333– 342 (1975).
4. Alroy, J. The Shifting Balance of Diversity Among Major Marine Animal Groups. 329, 1191–1194 (2010).
5. Close, R. A., Benson, R. B. J., Saupe, E. E., Clapham, M. E. & Butler, R. J. The spatial structure of Phanerozoic marine animal diversity. *Sci New York N Y* 368, 420–424 (2020)

Review 2

This paper represents an excellent contribution to the methods available for studying biodiversity in the fossil record. The DeepDive method is an important advance for our ability to circumvent biases that affect the scope of the sampling universe captured in the fossil record (geographic space, taphonomic biases, etc).

With existing methods, estimating global diversity would have been largely intractable, due to spatial bias in the fossil record. With DeepDive, however, we may now actually be able to estimate this parameter with some confidence, provided that the simulations used in the training step encompass the range of plausible scenarios that generated the empirical data.

I think that the paper should be published in Nature Communications, but there are some relatively minor changes that need to be made, and I have some queries and comments on specific parts of the text. I have also annotated the PDF of the MS with suggested edits and comments.

We thank the reviewer for taking the time to evaluate our proposal and for their constructive feedback, which we have carefully taken into consideration in revising our paper.

General questions:

- Can the authors comment on whether the model can do a good job of inferring biodiversity trajectories outside of those encountered in the training data? How do we know when the breadth of the simulated training data isn't adequate? Will widespread application of this method result in disagreements between camps of researchers who differ in the way they think biases should be simulated in the training data, yielding different resulting curves? How could this scenario be resolved?

In an ideal world, the model wouldn't be required to estimate outside the range of parameters used in the training data, that is, provided that the simulated training datasets span a sufficiently large range of settings. However, we acknowledge that it is virtually impossible to account for all possible scenarios in the training phase. To this end we had included the three datasets with pronounced temporal, taxonomic and spatial biases that are likely rare in the training set as they derive from simulations with different parameterisations (designed to enforce such strong biases). These tests showed that our model retained a good accuracy (Figure 2).

In our revisions, following the Reviewer's advice, we included two new testsets, this time simulated under models that effectively did not exist in the training data. The first model included multiple mass speciation and mass extinction events, creating a *spiky* diversity trajectory, while the second included diversity dependent

diversification regulated by a carrying capacity with two mass extinction events occurring at predefined times (*Results* lines 150-164, *Methods* lines 547-558). We found that the accuracy of our initial model was lower for these test sets compared with the other tests (Figure 2). However, we also demonstrate that accuracy can be improved substantially when these patterns are included in the training set (Supplementary Figure 8).

Indicators that the breadth of the training simulations are inadequate can include: parameters of empirical data that don't fall within the range of the training set (see histograms), if previously hypothesised patterns are particularly rare or absent from the training set, if key information e.g. about biogeography is missing from the training set. It is possible that different parameterisations may lead to disagreements, but the framework necessitates that assumptions are explicit and therefore can be discussed. The framework could potentially be used to assess the impact of different sets of assumptions about biases in the empirical data on predicted diversity curves. We add a comment on this to the discussion (lines 216-220).

- Can the authors comment on how the model could be used over time intervals when palaeogeographies change? Can changing region definitions (including appearances and disappearances of regions) be accommodated by the DeepDive model?

Changes in palaeogeography can be accommodated by the DeepDive model, and we demonstrate some aspects of this in the case of the custom proboscidean model as they disperse through time across regions that can only be reached after some event (e.g. the formation of the Isthmus of Panama). Additionally, the model provides high flexibility in e.g. allowing for regions to essentially merge by raising their connectivity to 1, or split apart by decreasing their connectivity through time. We now describe these options in the main text of our revised manuscript (lines 74-80).

Remarks and queries about particular sections of the text:

- 61–64: Are the inferred biodiversity trajectories biased or influenced by the `_distribution_` of trajectories recorded in the training data? I.e., will the model tend to infer the correct biodiversity trajectory if that kind of pattern is present but very rare in the training data? For example, if the vast majority of the simulated biodiversity histories are not diversity dependent, but a tiny fraction of them `_are_` diversity dependent, would the model do a good job of inferring diversity dependence if it was truly present in the empirical data?

Inference can be improved if certain patterns are more common in the training set. As described in our reply above, we now add two test sets representing patterns that are rare in the training set used in the manuscript (diversity dependence followed by mass extinction, mass speciations and mass extinctions) and demonstrated how accuracy changes when these patterns are explicitly contained in the training set for a new model (lines 149-164, Figure 2, Supplementary Figures 5, 7-8).

- 76–77: Do independently-generated validation and test datasets have the same generating model parameters? Would the performance of the model be worse if test datasets were generated under different model parameters? Or is this not much of an issue for assessing the model's performance, more generally?

We address this point with region, taxon and time bias test sets as these are generated with parameters differing from the training and validation sets. We now clarify in the text that these simulations were generated under a different parameterisation compared with the training set (lines 545-558).

- 91–92: How strongly does the model want to infer temporal autocorrelation in diversity trajectories? I.e., does it generally tend to smooth out short-term variation, as in the smooth trajectory of S3A (I realise that the true trajectory was also smooth)? Or can it also correctly infer repeated and sudden short-term changes in diversity? I'm just wondering if the smoothness of the trajectories in the training dataset would result in inferred trajectories for empirical datasets being quite smooth by default, even if the true trajectory wasn't. On a related note, is the size of the confidence interval envelope for the inferred trajectory larger across all intervening bins when the empirical record has large short-term fluctuations in sampling completeness, like in the examples of Fig. S3?

The two additional test sets added to the analysis should address this point, as they implement mass speciations and mass extinctions causing spiky diversity patterns. We find that the prediction error tends to be slightly higher following sudden changes of diversity, but also that the model can provide substantially better estimates of these changes when similar patterns are explicitly included within the training set (lines 545-558 for *Methods*, 149-164 for *Results*, Figure 2, Supplementary Figures 5, 7-8). We note that the variation in size of the confidence intervals in Figure S3 is more related to higher standing diversities than to short-term fluctuations in sampling completeness. This is also seen in our Proboscidea estimations where for higher diversity estimates we observe larger intervals. Uncertainty through time is more likely to be consistent when rescaled by standing diversity.

- Line 122 and elsewhere: Regarding the comparisons between DeepDive and SQS — I think that to be fair to SQS, and especially to prevent detractors from using your analysis as a reason not to use SQS in *_any_* context, it would be preferable to add some qualifications to these conclusions (as I mentioned in earlier comments on the PDF). SQS is a tool for standardising samples to equal sample completeness, which means that it's mainly geared towards correcting for uneven sampling intensity. It's not able to correct for variation in the size of the underlying sampling universe (spatial, taphonomic, taxonomic, etc.) — and neither can any method solely aimed at correcting for sampling intensity. DeepDive can obviously do both, but SQS is still an objective and reliable way to standardise diversity samples, given that certain assumptions are met. E.g., it would be good for standardising

alpha diversity within localities across samples, or within spatial regions if these regions are fairly comprehensively sampled. If sampling is random and unbiased, and the size of the sampling universe is controlled for, then SQS does a good job of telling you how many species are found, on average, within a random sample of $q\%$ of the individuals in a population (e.g., 60% of the individuals at a quorum of 0.6, or 90% at a quorum of 0.9). I think that's biologically meaningful and informative.

We agree and have now edited the discussion of SQS to reflect these points and clarify the difference in purpose of SQS (lines 209-212).

- 247: Are there limits to how severe spatial biases can be before DeepDive cannot reliably estimate global diversity, and would we be able to recognise this situation? E.g., how would DeepDive cope with sampling regimes where entire continental regions are absent from the global fossil record for protracted intervals of geological time? If you look at how regions are sampled through the Phanerozoic in tetrapods (Close et al. 2020 Proc B) you'll see that there is a very heavy bias towards Europe and North America, with many other regions entirely absent from the record for long intervals. It might be useful to comment on these limits of DeepDive in the discussion.

Within the simulation framework scenarios of regions being unsampled for periods of time or not at all can be specified, and models could be specifically trained to deal with these kinds of cases. To further assess this type of scenario, additional test sets could be designed that include regions appearing and disappearing. We think the ability of our simulation to flexibly reproduce different diversification and preservation patterns will allow users to determine case-by-case whether the model can generate robust estimates for their data.

- 311: Would you need to include (e.g.) diversity-dependent and early-burst models in the simulations in order to infer those patterns in empirical data? Even a wide range of diversity trajectories were generated in your simulations, are constant models with rate-shifts sufficient for all likely scenarios?

In our revision we now implemented these processes in our software so that mass speciation and diversity dependence can now be explicitly included in the simulations (see also replies above).

We address comments included in the annotated pdf. We have however decided to retain the original name of the software, DeepDive, as the repositories and documentation have already been developed under this name.

Review 3

Summary

This study uses synthetic data for training a deep learning model aiming to infer the geological history of the global scale species richness. Specifically, the authors generate abstracted fossils records consisting of species occurrences distributed over arbitrary time intervals and predefined geographic units. These abstracted records are generated using a birth-death process and later filtered using both sampling and preservation filters to resemble the physical fossil record. The abstracted fossil records are summarized using different features, which are used to train a bidirectional long short-term memory neural network to predict species richness over time. Authors claim the proposed approach outperforms alternative methods under a wide range of preservation scenarios. Overall, this work provide a great alternative to overcome data shortages for machine learning model training in paleobiology, specifically for estimating biosphere-scale species richness, but with potential to test a number of hypothesis in metazoan macroevolution. I have no major issues with the data, methods, and conclusions, and I recommend publication with some minor revisions. Below I provide some suggestions that may help to improve the manuscript.

We thank the Reviewer for their positive feedback and for raising the comments below, which we have carefully addressed in our revisions.

COMMENTS

On the simulated fossil records

Comment: I very much like the idea of using simulations to generate fossil records that capture biogeography structure. However, I would like to know why the authors assigned taxa to predefined bioregions to capture "biogeography of taxa". This approach may not reflect the biogeographic structure underlying the data. 1 Perhaps biogeographic structure can be extracted from the data without using a predefined number of discrete biogeographic regions. Recent studies show that unsupervised community detection on network representations of the same empirical data captures biogeography. So, I am wondering if it would possible to use network community structure to capture biogeographic signal in both physical (=empirical data) and abstracted records (=simulated data). For instance, to compute the effective number of modules/communities at a given time. After all, using a predefined number of discrete biogeographic regions that remains unchanged over time does not really capture temporal changes in spatial structure.

It would be possible to use a method such as network community detection or a minimum spanning tree and have now added a comment on the potential for this in the manuscript (lines 252-255). However, we opted for the use of discrete biogeographic regions mostly because of two reasons. First, this allows us to include interpretable areas for which a history of connectivity through time might be

well characterised from empirical data. For instance, the use of continents in the analysis of elephants allowed us to encode in the training settings an African origin, the Oligocene increase in connectivity to Eurasia and the effect of the formation of the Isthmus of Panama. We note however, that the areas utilised in the marine invertebrate analyses were in fact determined using a community-detection algorithm (minimum spanning tree) run in the original paper analysing this record (Flannery-Sutherland et al, 2022).

The second reason is efficiency of the simulation, which is necessary as we need to simulate thousands of datasets in reasonable time. Discrete biogeography provides a much more efficient way to approximate spatial distribution compared with continuous space simulations, such as those implemented in the gen3sis software (Hagen et al, 2021).

Finally, we emphasise that our biogeographic regions do change through time in the simulations used here: areas appear and disappear, connectivity changes, dispersal probability varies, and area-specific carrying capacities vary over time. Thus our simulations do include a range of dynamics in the spatial structure of the data, even though at coarse resolution.

On the empirical studies

Comment: When estimating diversity from empirical data, the authors presented a case study that seems to include benthic and planktonic marine clades all together. Perhaps it is not a surprise that the simulated trajectory do not capture CPE. To test for CPE, it would be better to restrict the analysis to the clades that has been observed to be affected by such an event, otherwise this result remains somewhat speculative.

Previous studies in the literature have shown that the CPE is the sum of relatively undynamic diversity losses across a range of clades. At first glance, gastropods appear to be an exception to this, but this is almost certainly a lagerstätte effect driven by extensive sampling of a well preserved and hyper-diverse Early Carnian assemblage in Italy called the San Cassian. Unpublished work on Bivalves using the regionalised PyRate approach of Flannery-Sutherland finds nothing noteworthy at the CPE. It is possible also that extinction dynamics are present at the species level rather than the genus level, e.g. conodont results in Figure 3 of Flannery-Sutherland et al, 2022. Ammonoids could be a potentially good candidate to investigate impacts of the CPE in order to avoid issues of polyphyly, and they do have marked family-level turnover at the event - though we consider this to be better suited to a dedicated study and is probably outside the scope of this paper.

Flannery-Sutherland's paper suggests that the CPE is regionally heterogeneous and manifests most strongly in the West Tethys, a further issue is that the West Tethyan

record has the most precise dating constraints for pre and post CPE occurrences. Perhaps the West Tethys could be a future region of focus then. We now provide a more detailed discussion around some of these points in our revised manuscript (lines 296-301).

TJME vs PTME Comment:

The authors indicate that "patterns of genus-level diversity loss occurring in the lead up to and at the TJME are broadly consistent with previous hypotheses on global biodiversity changes at this time". However, the pattern (in Figure 6) also seems to indicate that TJME is stronger than PTME. Is that also consistent with previous hypotheses? The study provides a global measure of how good are the simulated trajectories. I am wondering if would be possible to assess that at the stage and epoch level.

The exact magnitude of extinction events may be difficult to compare due to massive differences in sampling between the Late Triassic and Early Jurassic. It is possible DeepDive is not fully correcting this discrepancy. In our revisions, we ran analyses on a new test set where mass extinctions were simulated at fixed times and found that prediction error tends to be higher following sudden changes in diversity (Supplementary Figure 9, lines 158-164). Given that very sharp changes in diversity can lead to higher levels of uncertainty around mass extinction events, our ability to compare these events might be limited. We add a comment on this in the discussion (lines 296-312).

Does DeepDive "outperforms the most widely used current approaches"?

Comment: The study provides a global metric to evaluate the performance of the trained model. I am wondering if it would be possible to assess the performance at the stage and epoch levels that underlies the empirical data. We could learn a lot from that. For instance, what happens across known major geological events? Does the performance of the trained model drops? Does it recover gradually after? I believe that could be more interesting than the trajectory itself as it may capture major events over time and their relative impact on the species richness.

As mentioned in our replies above, we have addressed this point by adding a new testset simulated under predefined and fixed times of mass extinctions. This allowed us to measure the prediction error through time and to assess the effect of sudden diversity changes on the accuracy. We found that indeed there is an increase in prediction error following mass extinctions, which however quickly returns to lower values after that (Supplementary Figure 9, lines 545-558 for *Methods*, 149-164 for *Results*). The error decreases when mass extinctions are better represented in the training data, as we show based on a model optimised on a new training set.

I hope you found these ideas useful,

- Alexis Rojas

Very useful indeed, many thanks for the insightful comments.

Reviewers' Comments:

Reviewer #1:

Remarks to the Author:

I thank the authors for their effort in incorporating and discussing all my suggestions. I think the manuscript is in great shape and I have no further comments.

Reviewer #2:

Remarks to the Author:

General comments

- Overall I am very happy with the substantial efforts made by the authors to address my comments and suggestions, and those of the other reviewers.
- In the Discussion paragraph beginning on L212, I think it's worth reiterating how important it is that the true diversification scenario is included in the training dataset, and how the predictions suffer if those true scenarios are absent or rare. I acknowledge you do this already to a reasonable extent around L241. However, I believe that it's worth explicitly re-stating that the performance of DeepDive is otherwise only as good as SQS in such scenarios (as shown by Figure 2).

Minor issues

- L85 (and possibly elsewhere): remove contractions like "you'd"  "you would" (or rephrase to passive voice in this instance — e.g., rephrasing to "The distribution of parameters in the simulated datasets can be compared to those in the empirical occurrence data, to ensure the range of parameters that are expected based on the empirical data fall within the range the model has had the opportunity to learn from.")
- L212: "Indeed, although methods like SQS are widely used for standardising diversity estimates, their scope is to standardise samples to equal completeness, while they are not designed to estimate global diversity." Suggest rewording to this: "Indeed, although methods like SQS are widely used for standardising diversity estimates, their intended purpose is to standardise samples to equal sampling intensity or completeness; they are not designed to control for variation in the scope of the accessible sampling universe, which is of central importance when estimating global diversity."
- Figure 4: The caption states that "Parameters which describe the dataset are contained within the parameters of the simulations." Yet I note that the empirical (orange) distributions often look quite different to those of the simulated (blue) distributions, especially for panels A, E and F. Is this a problem at all? (I.e., if the frequency distributions of parameters differ even if the ranges are comparable?)

-- Roger Close

Reviewer #3:

Remarks to the Author:

Review of Cooper et al.

Summary

I'm satisfied with how the authors addressed my first review. The revised manuscript is a great read. I recommend accepting it with just a very minor revision.

[LINE 252] Bioregions could be informed by using methods such as network community detection (56) or minimum spanning trees, e.g. (24, 29), to ensure they reflect the biogeographic structure of the data.

Comment: I would suggest adding the reference Vilhena and Antonelli since it appears to be the first study using community detection to outline bioregions.

- Alexis Rojas

References

Daril A. Vilhena and Alexandre Antonelli. A network approach for identifying and delimiting biogeographical regions. 6:6848. ISSN 2041-1723. doi: 10.1038/ncomms7848. URL <http://www.nature.com/doifinder/10.1038/ncomms7848>.

Reviewer #4:

Remarks to the Author:

This is the first time I have seen the manuscript, so my ideas are going to be new to the previous revisions that you received. I would also be really interested to see a long-format presentation on the research to compliment the article, so I can better understand what was done here. But I should also be clear, I thought this study was really novel, interesting and made me think a lot – the following comments are just to help the authors think about reframing aspects to make it more understandable to a wider readership.

I guess my important question is, what explicitly is the DeepDive model? I kept getting confused, if it was the whole protocol or just the RNN model. In the methods, it just turns up undefined in line 534. From my own experience and from looking at the code (it is great that you made this publicly available!), it looks like the RNN is re-set up each time you run the code (not a problem), but it does mean that every time you run the ML algorithm that the model and results will always be different – the nodes will connect in different ways each time. So, if DeepDive refers to the ML algorithm then I would say you cannot name it, because it is always changing, but if it relates to the protocol then you should rename it the DeepDive protocol and not model, just for clarity.

One aspect you have to overcome with data science is that it doesn't matter how good the methods are, the data needs to be good too (unless it is a thought experiment or just to establish a protocol):

*For this study, I am left uncertain if data, such as singletons, is good data. We know singletons = poor data, which is why they are often removed from previous studies of richness estimation. So why is that data informative here? If anything, it must compromise the model because what information can you get from singletons?

*It looks to me that the data fed into the RNN algorithm (training dataset) is always tabular data. So why would you use a neural network approach instead of a classification tree approach. NNs are normally for noisy data, like audio and images. It is also interesting because an RNN model is virtually impossible to understand (black box), but a classification tree approach is interpretable and maybe you could have got some insights into what determines the biodiversity trajectories had you used this approach.

*NNs require a lot of data to be trained on and they are only as good as the data that they are trained on, because they have the tendency to overfit. In the same vein, would a decision tree approach have been better here?

*I think it is a bit dishonest to self-cite yourselves for saying where the dataset came from, especially when you downloaded it from the paleobiology database. I also have concerns about the quality of the "Song" database, because it is not available for public scrutiny, e.g., the references where the data originally came from.

I think regarding these points you just need to add some information to demonstrate that your data is good data and that a deep learning approach was the right approach compared to the alternatives (I do not see how Figs.S4-7, as written in the text, show that using a NN was the best approach).

Minor comments:

In the abstract you say your method "outperforms" alternative approaches. That is untrue, because you will never know, until you know what the right answer actually is. You know, that you don't know what actually happened, you are just hypothesising based on your results. Birth-death models and neural networks are very popular methods at the moment, but it is just a different way of looking at things and not necessarily more accurately and unlikely with more precision. Personally, I would reword to a "new perspective".

Line 41-43. I think you need to make the text a bit simpler. How many people actually know what deep learning is? I have learnt through the review process that most palaeontologists have a really poor understanding of machine learning algorithms.

Line 43 vs. Line 46. You have DeepDive model as singular and then DeepDive models as plural. It is

unclear what you mean here. Do you train one model and then apply it to datasets or is DeepDive a methodological protocol? At this point of the manuscript the reader is left confused.

Line 46-48: Late Permian to Early Jurassic is not an animal group, what you mean is two datasets.

Line 263-265: The cited references did not use neural network classifiers. Tietje & Rödel used a randomforest algorithm, Foster et al. used Xboost (gradient boosted trees and then game theory to understand how the model made its predictions) and Finnegan et al. used a gradient boosted trees algorithm. I think you should expand the sentence to be clearer about what the previous studies did and how yours is different. I think you should also add Raja et al. (2021) <https://doi.org/10.1111/geb.13321> which actually used a deep learning algorithm.

Then there is also the other type of machine learning applications in paleobiology/macroevolution you have not cited, like Edie et al. (2023) *Frontiers* and Tetard et al. (2020) *Climate of the Past* that use NN to do taxonomy from images.

I think any comments about the Carnian Pluvial event should be deleted. Your analysis is too coarse to see the impact of it. The same way you wouldn't see the P-Tr event had the Induan been merged with the Olenekian.

Finally, my understanding is that (1) you simulated diversity using birth-death models, which were then (2) degraded using the fossilisation and preservation simulator, this was then used to (3) train and evaluate the performance of an RNN model. This trained model was then (4) applied to the empirical datasets to make predictions about how diversity changed overtime. If this is correct, then Fig. 1 is actually confusing, and you should split the figure into step 1 (model setup and evaluation) and step 2 (model application).

REVIEWER COMMENTS

Reviewer #1 (Remarks to the Author):

I thank the authors for their effort in incorporating and discussing all my suggestions. I think the manuscript is in great shape and I have no further comments.

We thank the reviewer for taking the time to review our paper.

Reviewer #2 (Remarks to the Author):

General comments

- Overall I am very happy with the substantial efforts made by the authors to address my comments and suggestions, and those of the other reviewers.
- In the Discussion paragraph beginning on L212, I think it's worth reiterating how important it is that the true diversification scenario is included in the training dataset, and how the predictions suffer if those true scenarios are absent or rare. I acknowledge you do this already to a reasonable extent around L241. However, I believe that it's worth explicitly restating that the performance of DeepDive is otherwise only as good as SQS in such scenarios (as shown by Figure 2).

Many thanks for taking the time to review the paper, and for the detailed comments. We now reiterate the importance of the diversification scenario falling within the scope of the training data on lines 231-235.

Minor issues

- L85 (and possibly elsewhere): remove contractions like "you'd"  "you would" (or rephrase to passive voice in this instance – e.g., rephrasing to "The distribution of parameters in the simulated datasets can be compared to those in the empirical occurrence data, to ensure the range of parameters that are expected based on the empirical data fall within the range the model has had the opportunity to learn from."

The statement on line 85 has been rephrased as suggested and contractions have been removed from the text.

- L212: "Indeed, although methods like SQS are widely used for standardising diversity estimates, their scope is to standardise samples to equal completeness, while they are not designed to estimate global diversity." Suggest rewording to this: "Indeed, although methods like SQS are widely used for standardising diversity estimates, their intended purpose is to standardise samples to equal sampling intensity or completeness; they are not designed to control for variation in the scope of the accessible sampling universe, which is of central importance when estimating global diversity."

This sentence has been rephrased as suggested [lines 212-216].

- Figure 4: The captions states that "Parameters which describe the dataset are contained within the parameters of the simulations." Yet I note that the empirical (orange) distributions often look quite different to those of the simulated (blue) distributions, especially for panels A, E and F. Is this a problem at all? (I.e., if the frequency distributions of parameters differ even if the ranges are comparable?)

The distributions are indeed different but result in the case of the empirical data from a single dataset, while they summarize 1000 data sets in the case of the simulations. Deep learning models can be expected to misbehave when used to extrapolate values outside of the training range. The histograms show that the range of values of the empirical features is firmly within the range of simulated values, as we now express more clearly in the Figure captions of Figures 4 and 5.

-- Roger Close

Reviewer #3 (Remarks to the Author):

I'm satisfied with how the authors addressed my first review. The revised manuscript is a great read. I recommend accepting it with just a very minor revision.

[From attachment]

Review of Cooper et al. Summary I'm satisfied with how the authors addressed my first review. The revised manuscript is a great read. I recommend accepting it with just a very minor revision. [LINE 252] Bioregions could be informed by using methods such as network community detection (56) or minimum spanning trees, e.g. (24, 29), to ensure they reflect the biogeographic structure of the data. Comment: I would suggest adding the reference Vilhena and Antonelli since it appears to be the first study using community detection to outline bioregions. - Alexis Rojas References Daril A. Vilhena and Alexandre Antonelli. A network approach for identifying and delimiting biogeographical regions. 6:6848. ISSN 2041-1723. doi: 10.1038/ncomms7848. URL <http://www.nature.com/doifinder/10.1038/ncomms7848>.

We thank the reviewer for taking the time to comment on the paper and have added the suggested reference [line 262].

Reviewer #4 (Remarks to the Author):

This is the first time I have seen the manuscript, so my ideas are going to be new to the previous revisions that you received. I would also be really interested to see a long-format presentation on the research to compliment the article, so I can better understand what was done here. But I should also be clear, I thought this study was really novel, interesting and made me think a lot – the following comments are just to help the authors think about reframing aspects to make it more understandable to a wider readership.

I guess my important question is, what explicitly is the DeepDive model? I kept getting confused, if it was the whole protocol or just the RNN model. In the methods, it just turns up undefined in line 534. From my own experience and from looking at the code (it is great that you made this publicly available!), it looks like the RNN is re-set up each time you run the code (not a problem), but it does mean that every time you run the ML algorithm that the model and results will always be different – the nodes will connect in different ways each time. So, if DeepDive refers to the ML algorithm then I would say you cannot name it, because it is always changing, but if it relates to the protocol then you should rename it the DeepDive protocol and not model, just for clarity.

We thank the reviewer for their interesting comments on the manuscript and the very swift response. We agree the definition of what DeepDive is can be made clearer.

DeepDive is a modular implementation of 1) a simulation tool, generating biodiversity data in space and time using birth-death and dispersal processes, and a fossil data based preservation processes with sampling biases, and 2) a deep learning tool, allowing the user to setup and train predictive models based on RNNs and to use them make inferences from empirical fossil data. DeepDive is also the name of the Python library we developed to perform all these tasks. We now clarify this definition in the revised text and changed references to the DeepDive as a 'model' [lines 43, 46, 92, 121, 142, 168, 318, 573-574, captions of Figure 2, Supplementary Figures 6-8, Supplementary Table 2].

One aspect you have to overcome with data science is that it doesn't matter how good the methods are, the data needs to be good too (unless it is a thought experiment or just to establish a protocol):

*For this study, I am left uncertain if data, such as singletons, is good data. We know singletons = poor data, which is why they are often removed from previous studies of richness estimation. So why is that data informative here? If anything, it must compromise the model because what information can you get from singletons?

We agree, and this is why we tested our trained models against a set of simulations where the data were degraded in multiple ways, including strong temporal, spatial, and taxonomic biases. This allowed us to test our model (and the widely used SQS model for comparison) against a range of datasets specifically generated to be of lower quality. Our

analyses indicated that our deep learning approach outperformed the alternative method under most circumstances.

We opted to retain singletons in our analyses as they have previously been used to inform coverage-based methods for estimating past biodiversity, such as Good's u , Chao1, λ^5 (see e.g. Close et al. 2018. How should we estimate diversity in the fossil record? Testing richness estimators using sampling-standardized discovery curves. *Methods in Ecology and Evolution*). Using singletons as a feature from simulations in the training set provides the neural network with opportunity to learn how this pattern of sampling, that is frequent in fossil occurrence datasets, impacts inference. The model can deal with input data where there are singletons so long as the frequency of these in the empirical data falls within the range the training data covers. The model should be able to make inferences without discarding data (which we hope will be useful for analysing more depauperate taxa).

*It looks to me that the data fed into the RNN algorithm (training dataset) is always tabular data. So why would you use a neural network approach instead of a classification tree approach. NNs are normally for noisy data, like audio and images. It is also interesting because an RNN model is virtually impossible to understand (black box), but a classification tree approach is interpretable and maybe you could have got some insights into what determines the biodiversity trajectories had you used this approach.

We decided to use RNNs in the development of DeepDive as they are specifically designed to analyse time series data. While we agree that classification or regression trees provide more interpretable parameterizations, they do not provide a direct way to incorporate the temporal dimension that characterizes our input data. The data are indeed tabular but they result from stacking multiple time series (e.g. providing number of occurrences, localities, sampled species, etc. per time bin). Recurrent neural networks are specifically designed to learn from a sequence (in this case a time series) of features and we therefore think they are best suited for modeling data in the context of palaeodiversity through time. Yet, we agree that alternative models can provide valuable alternative options and in designing our software we developed it in a modular form that would easily allow a user to simulate a training dataset using DeepDive and then use it to train a different model, for instance based on random forests using standard Python libraries for machine learning such as `sklearn`. We now provide additional justification for our model choice and clarify that other models can easily be implemented based on DeepDive simulations [lines 279-284].

*NNs require a lot of data to be trained on and they are only as good as the data that they are trained on, because they have the tendency to overfit. In the same vein, would a decision tree approach have been better here?

We agree, while also noting that NNs (and indeed most models) can be only as good as the data they are fed with. We chose to use a deep neural networks for their ability to approximate virtually any function and because they can account for the complex dependencies among time bins, species richness, and its spatial distribution. To prevent over-fitting, we followed standard machine learning practices using a stopping rule during the training based on the loss calculated on a validation set [lines 525-528]. As mentioned above and in the revised manuscript [lines 279-284] our DeepDive library easily allows a user to train other models using DeepDive-generated simulations.

*I think it is a bit dishonest to self-cite yourselves for saying where the dataset came from, especially when you downloaded it from the paleobiology database. I also have concerns about the quality of the “Song” database, because it is not available for public scrutiny, e.g., the references where the data originally came from.

I think regarding these points you just need to add some information to demonstrate that your data is good data and that a deep learning approach was the right approach compared to the alternatives (I do not see how Figs.S4-7, as written in the text, show that using a NN was the best approach).

We note that both our datasets were not directly downloaded from the original databases instead obtaining them from the publicly available datasets that were published alongside Flannery-Sutherland et al. (2022) and Cantalapiedra et al. (2021) respectively. These datasets result from cleaning steps, e.g. duplicate removals and taxonomic standardization, described in the original papers, which is why in both cases we cite the studies that made these cleaned datasets available. Yet, we now additionally cite in our revised text the original sources of the datasets (lines 569-573).

For the Song data, a formal assessment of data quality would be somewhat subjective and outside of the scope of this study, but its inclusion of substage dating and the more consistent spatiotemporal coverage compared to the PBDB speak to its quality. Song’s reference list is publicly available for scrutiny here:

Database S1. Late Permian-Triassic fossil database

Song, H., Huang, S., Jia, E., Dai, X., Wignall, P., Dunhill, A. (2020). Data from: Flat latitudinal diversity gradient caused by the Permo-Triassic mass extinction. Dryad.

<https://doi.org/10.5061/dryad.41ns1rn9z>

We show that the method is better than the bench mark (SQS) at reducing errors in these figures. We have updated Figure 2 to include plots that were previously supplementary to illustrate the reduced error rate. In S4 the panels illustrate the variation in relative MSE across different levels of completeness, preservation rate, number of sampled species, whether a clade is extinct of extant, the duration of species and the duration of clades

when estimates are made for test sets using DeepDive. S5 shows the results for the same data but with analysis performed with SQS. In comparing the two figures we observe, for example, the distribution of rMSE scores is wider and higher across different levels of completeness in SQS estimates than in those made using DeepDive – these differences are discussed in the text [lines 125-137, 146-150, 228-231]. S6 (previously S7) demonstrates that DeepDive estimates can be improved when patterns are better represented in the training data with rMSE for DeepDive estimates over an order of magnitude lower than rMSE for the same test sets using SQS [lines 231-235]. We now clarify that other machine learning models could be used to replace the deep neural networks used here thanks to the modular structure of our software [lines 279-284].

Minor comments:

In the abstract you say your method “outperforms” alternative approaches. That is untrue, because you will never know, until you know what the right answer actually is. You know, that you don’t know what actually happened, you are just hypothesising based on your results. Birth-death models and neural networks are very popular methods at the moment, but it is just a different way of looking at things and not necessarily more accurately and unlikely with more precision. Personally, I would reword to a “new perspective”.

We make sure that in the revised text we only use the word outperforming when referring to simulation-based testing, that is when a ground truth is indeed known [abstract, lines 144, 226, 300].

Line 41-43. I think you need to make the text a bit simpler. How many people actually know what deep learning is? I have learnt through the review process that most palaeontologists have a really poor understanding of machine learning algorithms.

We agree with the reviewer that the specific meaning of ‘deep learning’ might be only partly understood by part of the readership. At the same time, the use of the term has become widespread across not only scientific publications but in popular science and news articles as well. It is also the accurate term to define the type of models used here and we therefore preferred to maintain the current version of the text. We do refer to general machine learning papers in several places throughout our manuscript where readers can find more information about these methods.

Line 43 vs. Line 46. You have DeepDive model as singular and then DeepDive models as plural. It is unclear what you mean here. Do you train one model and then apply it to datasets or is DeepDive a methodological protocol? At this point of the manuscript the reader is left confused.

We rephrased this part and now refer to the DeepDive approach and DeepDive trained models [lines 43, 46].

Line 46-48: Late Permian to Early Jurassic is not an animal group, what you mean is two datasets.

We rephrased this part to: “We then use DeepDive trained models to estimate global biodiversity dynamics for two animal groups: marine animals from the Late Permian to Early Jurassic (24) and the mammalian clade Proboscidea (40).” [line 47]

Line 263-265: The cited references did not use neural network classifiers. Tietje & Rödel used a randomforest algorithm, Foster et al. used Xboost (gradient boosted trees and then game theory to understand how the model made its predictions) and Finnegan et al. used a gradient boosted trees algorithm. I think you should expand the sentence to be clearer about what the previous studies did and how yours is different. I think you should also add Raja et al. (2021) <https://doi.org/10.1111/geb.13321> which actually used a deep learning algorithm.

We add comment that these methods used gradient boosted trees and random forests. The citation to Raja et al. (2021) has been added [line 272-275].

Then there is also the other type of machine learning applications in paleobiology/macroevolution you have not cited, like Edie et al. (2023) *Frontiers* and Tetard et al. (2020) *Climate of the Past* that use NN to do taxonomy from images.

We add a comment that neural networks are also being used in taxonomy and morphological studies, citing Edie et al. (2023) and Tetard et al. (2020) as well as He et al. (2024) [line 276-277].

I think any comments about the Carnian Pluvial event should be deleted. Your analysis is too coarse to see the impact of it. The same way you wouldn't see the P-Tr event had the Induan been merged with the Olenekian.

We agree and remove references to the CPE from the manuscript.

Finally, my understanding is that (1) you simulated diversity using birth-death models, which were then (2) degraded using the fossilisation and preservation simulator, this was then used to (3) train and evaluate the performance of an RNN model. This trained model was then (4) applied to the empirical datasets to make predictions about how diversity changed overtime. If this is correct, then Fig. 1 is actually confusing, and you should split the figure into step 1 (model setup and evaluation) and step 2 (model application).

We thank the reviewer again for taking the time to comment on the manuscript. The figure headings have been updated to clarify the structure between “Mechanistic simulations and model training” and “Empirical predictions.”

Reviewers' Comments:

Reviewer #4:

Remarks to the Author:

I have gone through the response to reviewers and the revised manuscript and the authors have occasionally made cursory changes to address my comments, but mostly just explained my comments/questions (sometimes inadequately, for example there are many examples of non-deep learning techniques for handling time series analyses in biomedical studies) without making any changes. Even though I find that disappointing, as I hoped the comments would have been genuinely useful for improving the robustness, novelty and communication of their study. That said, this manuscript is good enough for publication in this journal and the project is nonetheless interesting, so I will refrain from making any new comments.

The thing I like most about this project is that it is another example of how machine learning can be used in a novel way to investigate a big question in paleontology.

REVIEWERS' COMMENTS

Reviewer #4 (Remarks to the Author):

I have gone through the response to reviewers and the revised manuscript and the authors have occasionally made cursory changes to address my comments, but mostly just explained my comments/questions (sometimes inadequately, for example there are many examples of non-deep learning techniques for handling time series analyses in biomedical studies) without making any changes. Even though I find that disappointing, as I hoped the comments would have been genuinely useful for improving the robustness, novelty and communication of their study. That said, this manuscript is good enough for publication in this journal and the project is nonetheless interesting, so I will refrain from making any new comments.

The thing I like most about this project is that it is another example of how machine learning can be used in a novel way to investigate a big question in paleontology.

In our previous submission, we had followed the reviewer's advice and clarified what the DeepDive approach represented and its modular structure (simulations and model training, and empirical predictions). Further we added references and additional discussion on different models beyond deep learning approaches that could be used within the DeepDive framework.

We have now made additional efforts to address the points of reviewer 4. We more explicitly give mention to non-deep learning models as alternatives to analyze time series data [lines 284-286]. However, we consider a full comparison of statistical and machine learning models for time series analysis as beyond the scope of our paper.

We added a reference to provide full account of the original sources of the empirical datasets analyzed in the study, following advice of the reviewer [lines 572-573].

Reviewer #4 (Remarks on code availability):

The code could have been published in a more user-friendly way. I am a big fan of Jupyter notebooks. But a version of the code should also be uploaded to a repository (e.g., Zenodo) as github is a dynamic code source.

As requested by the reviewer, we have published our code in a permanent repository hosted at zenodo.org (DOI: 10.5281/zenodo.10979237). We opted to provide the code as executable scripts as this will facilitate their use on a computing cluster.

We think these final adjustments improve potential reception of the study by the readership of Nature Communications.